# LLMs for Generalizable Language-Conditioned Policy Learning under Minimal Data Requirements

## Abstract

To develop autonomous agents capable of executing complex, multi-step decision-making tasks as specified by humans in natural language, existing reinforcement learning approaches typically require expensive labeled datasets or access to real-time experimentation. Moreover, conventional methods often face difficulties in generalizing to unseen goals and states, thereby limiting their practical applicability. This paper presents *TEDUO*, a novel training pipeline for offline language-conditioned policy learning. *TEDUO* operates on easy-to-obtain, unlabeled datasets and is suited for the so-called in-the-wild evaluation, wherein the agent encounters previously unseen goals and states. To address the challenges posed by such data and evaluation settings, our method leverages the prior knowledge and instruction-following capabilities of large language models (LLMs) to enhance the fidelity of pre-collected offline data and enable flexible generalization to new goals and states. Empirical results demonstrate that the dual role of LLMs in our framework—as data enhancers and generalizers—facilitates both effective and data-efficient learning of generalizable language-conditioned policies.

## 1 Introduction

**Motivation.** A central aim of AI research is to develop autonomous agents capable of solving complex, multi-step decision-making tasks based on human-provided instructions articulated in natural language. Current approaches, which rely on traditional reinforcement learning (RL) methods, require either vast amounts of data in the form of offline expert demonstrations or data collected from real-time interactions with the environment. Such data is expensive to gather and online experimentation may be impractical in many real-world scenarios. Moreover, even with substantial data, RL agents are often constrained to a limited set of previously attained goals, or their performance deteriorates significantly when tested on new goal-reaching tasks (Yang et al., 2023).

**Problem setting.** Given the above, this paper approaches the problem of learning generalizable language-conditioned policies under minimal data requirements. Specifically, we consider a fully offline setup with access to a pre-collected dataset of state-action transitions, $\mathcal{D}$, alongside an unpaired set of natural language commands, $\mathcal{G}^{tr}$. The dataset $\mathcal{D}$ consists of triplets $(x, a, x')$, where $x$ and $x'$ belong to a high-dimensional state space $\mathcal{X}$, and $a$ represents actions within an action space $\mathcal{A}$. The natural language goals $g \in \mathcal{G}^{tr}$ describe a subset of tasks achievable within the environment. We posit that such data is often easy to obtain by simply recording agents interact with the environment and creating a list of natural language commands corresponding to the tasks typically performed within that environment. For instance, in household robotics, $\mathcal{D}$ may be derived from random exploration of the environment, while $\mathcal{G}^{tr}$ may describe tasks such as "Go and open the window" or "Fetch me a cup of tea." In the case of personal assistants, $\mathcal{D}$ might be collected by recording the daily activities of a human interacting with their mobile device, while $\mathcal{G}^{tr}$ would describe goals like "Book a restaurant for 7 PM" or "Send an email to John." With no assumptions regarding how the offline data has been collected, nor access to the ground-truth state-transition dynamics or rewards, our aim is to learn a language-conditioned policy, $\pi^*$, that can determine optimal actions for previously unseen goals $g \notin \mathcal{G}^{tr}$ and states $x \notin \mathcal{D}$. For a formal definition of the problem setup, please refer to section 2.

Figure 1: *Overview of TEDUO.* ① The unlabeled dataset of state-action transitions is pre-processed with LLM-automated hindsight labeling and state abstraction. ② The resulting labeled dataset of abstract state transitions is used as the input to an offline RL algorithm to learn the optimal goal-conditioned policies for the finite set of training goals. ③ Knowledge of the optimal actions for each observed training goal is distilled into a base LLM via SFT. The fine-tuned LLM acts as a language conditioned policy generalizing to previously unseen states and language commands.

**Challenges.** The task of learning $\pi^*$ from $\mathcal{D}$ and $\mathcal{G}^{tr}$ alone might seem impossible without resorting to human supervision. We can immediately identify the following challenges: **C1) Unlabeled data.** The dataset $\mathcal{D}$ lacks explicit labels linking states $x \in \mathcal{X}$ to the goals $g \in \mathcal{G}^{tr}$. Nor does it include any rewards indicating the optimality of actions in relation to these goals. **C2) Limited exploration.** We are in an offline setup with our knowledge of the environment dynamics being constrained to the state-action transitions observed in $\mathcal{D}$. **C3) Unknown data collection policy.** We make no assumptions regarding the optimality of the data collection policy concerning the training or testing goals. The actions in $\mathcal{D}$ could be entirely random or generated by policies aimed at solving goals with an unknown relationship to those in $\mathcal{G}^{tr}$. **C4) Generalization to new goals and states.** Beyond solving goals from $\mathcal{G}^{tr}$ and taking optimal actions in previously observed states $x \in \mathcal{D}$, we want our agent to generalize to new states and language commands corresponding to novel goal states.

**Proposed Solution: LLMs to elevate conventional RL.** Recent advances in LLMs offer a promising solution to these challenges. LLMs, pre-trained on vast amounts of Internet data, possess the requisite prior knowledge to understand natural language commands and follow simple instructions. However, while LLMs excel at general language comprehension, their ungrounded knowledge is insufficient for executing complex, multi-step decision-making tasks in dynamic environments (Finn, 2024; Szot et al., 2024). In this paper, we propose a novel training pipeline for offline language-conditioned policy learning—**TEDUO**: Teaching the Environment Dynamics from Unlabeled Observations. TEDUO distills knowledge of the environment dynamics into a pre-trained LLM through supervised fine-tuning. This knowledge is obtained by learning optimal policies with traditional RL, based on the offline dataset augmented with LLM-generated state abstractions and labels. Thus, within TEDUO, LLMs fulfill the dual role of *cheap data enhancers* and *flexible generalizers*, elevating conventional RL to address challenges C1-C4.

**Contributions. 1)** We introduce TEDUO—a novel LLM fine-tuning pipeline, which, to the best of our knowledge, is the first to enable the learning of generalizable language-conditioned policies based solely on an unlabeled dataset of state-action transitions and an unpaired set of natural language commands. **2)** We demonstrate that LLMs can be effectively employed both as cost-effective data enhancers—automating critical tasks typically handled by humans—and as versatile generalizers—acting as a general language-conditioned policy capable of solving new tasks in previously unseen scenarios. **3)** We empirically show that TEDUO enhances both the data efficiency and generalization capacity of offline training, outperforming competing approaches by a significant margin. We analyze the scalability of our method and provide insights into the learning process, showing that fine-tuned LLMs acquire core skills, rather than simply memorize optimal actions.

## 2 PROBLEM FORMALISM

**Inputs.** We are given a dataset $\mathcal{D}$ of past interactions of an agent acting according to a data collection policy $\pi^\beta$. This dataset is represented as a collection of trajectories:

$$\mathcal{D} = \{\tau_i\}_{i \in \mathcal{I}}, \tau_i = \{(x_t, a_t, x_{t+1})\}_{t=0}^{T_i}, \quad x_0 \sim \rho, \; x_{t+1} \sim P(\cdot|x_t, a_t), \; a_t \sim \pi^\beta(\cdot|x_t), \quad (1)$$

where $P$ is the state transition function determining the next state given an action $a_t \in \mathcal{A}$ and state $x_t \in \mathcal{X}$ and $\rho$ represents a distribution of initial states. Alongside $\mathcal{D}$, we are provided with an unpaired set of training goals $\mathcal{G}^{tr}$ describing a subset of tasks an agent may attempt to solve within the environment. Each goal $g$ is expressed in a goal representations space $\mathcal{G}$. In this paper, we focus on learning language-conditioned policies, taking $\mathcal{G}$ to be the space of natural language.

**Modeling assumptions.** Denoting by $\mathcal{P}(\mathcal{X})$ the powerset of $\mathcal{X}$, we assume there exists a ground-truth mapping $\phi : \mathcal{G} \to \mathcal{P}(\mathcal{X})$ associating each goal $g$ with a subset of the state space, $\phi(g) = \mathcal{X}_g \subseteq \mathcal{X}$. We say that $g$ is achieved at time step $t$, if $x_t$ lies in $\mathcal{X}_g$.[1]. Then, the cumulative discounted reward: $\sum_{t=0}^{\infty} \gamma^t R_\phi(x_t, a_t, x_{t+1}; g)$, with $R_\phi(x_t, a_t, x_{t+1}; g) = \mathbb{1}\{x_{t+1} \in \phi(g)\}$ measures the optimality of actions taken by an agent with respect to achieving the goal $g$, where $\gamma \in [0, 1)$ is the discount factor penalizing long sequences of actions. We make no assumptions regarding the optimality of the data collection policy $\pi^\beta$ with respect to $\mathcal{G}^{tr}$ and thus, in what follows, we will view our pre-collected data as an un-ordered collection of state-action-state transitions, in short denoted as $\mathcal{D} = \{(x, a, x')\}$. We also do not assume access to either of the ground-truth state-transition dynamics $P$ or the goal-to-state mapping $\phi$, and consequently the reward $R_\phi$. We only require that $\mathcal{G}^{tr}$ contains goals corresponding to states that have been visited in $\mathcal{D}$ [2].

**The goal.** Given $\mathcal{D}$ and $\mathcal{G}^{tr}$, our objective is to learn a language-conditioned policy $\pi^*$, where

$$\pi^* = \arg\max_\pi \mathbb{E}_{g \in \mathcal{G}, x_0 \sim \rho} \sum_{t=0}^{\infty} \gamma^t \mathbb{E}_{a_t \sim \pi(\cdot|x_t;g), x_{t+1} \sim P(\cdot|x_t, a_t)} \left[ \mathbb{1}(x_{t+1} \in \phi(g)) \right] \tag{2}$$

Crucially, $\pi^*$ should generalize to novel goals $g \notin \mathcal{G}^{tr}$ and previously unseen states $x \notin \mathcal{D}$. We also require that $\pi^*$ not only generalizes to synonymous language commands, but also to goals corresponding to new goal-states, emphasizing our focus on evaluation in the wild.

## 3 THE METHOD: TEDUO

To address the problem of learning language-conditioned policies solely based on the inputs $\mathcal{D}$ and $\mathcal{G}^{tr}$, we must overcome the challenges C1-C4 outlined in the introduction. While conventional RL methods are successful at learning optimal policies within well-explored environments, they typically require additional data labeling and are limited in generalization to new, previously unseen language commands and states. In contrast, although LLMs can understand the meaning of sentences in natural language describing each goal, their skills lack grounding in relation to the environment's dynamics. Our pipeline, TEDUO, employs LLMs to enhance conventional RL, effectively addressing challenges C1-C4. TEDUO consists of three main steps:

Step 1.
LLMs as data enhancers
**Construction of abstract MDPs.** For each goal, $g \in \mathcal{G}^{tr}$, we construct an abstract MDP by employing LLM-automated hindsight labeling and state abstraction, addressing C1 and C2, respectively.

Step 2.
**Goal-conditioned policy learning with offline RL** After obtaining a labeled dataset for each goal in $\mathcal{G}^{tr}$, we solve the set of abstract MDPs using an out-of-the-box offline RL algorithm. As a result, we obtain a set of learned policies $\{\pi_g : g \in \mathcal{G}^{tr}\}$. The learned policies improve on naive imitation learning, addressing C3.

Step 3.
LLMs as generalizers
**LLM supervised fine-tuning.** We distill the knowledge about the environment dynamics and optimal actions into a pre-trained LLM with supervised instruction fine-tuning (SFT). This step grounds the prior knowledge of the base LLM in the environment dynamics, thus enabling generalization to new, previously unseen states and goals, addressing both challenges C2 and C4.

In the following paragraphs we explain in detail individual steps of TEDUO.

---

[1] In this paper, we focus on simple goals representable as a subset of the state space. This definition can easily be extended to more complex goals using temporal logic. We leave such extensions for future work.

[2] In practice, $\mathcal{G}^{tr}$ can consist of a much larger set of training goals. This set will be effectively reduced to the set of visited goals after the first step of our training pipeline.

## 3.1 STEP 1. CONSTRUCTION OF ABSTRACT MDPS

Starting with the dataset of unlabeled observations $\mathcal{D}$ and the training goals $\mathcal{G}^{tr}$, we first construct a set of abstract MDPs $\{\mathcal{M}^g : g \in \mathcal{G}^{tr}\}$, where $\mathcal{M}^g := (\mathcal{S}^g, \mathcal{A}, P^g, R^g, \rho, \gamma)$, with $\mathcal{S}^g$ being the abstract state space for the goal $g$, $P^g$ the induced transition operator, and $R^g$ the reward function with respect to the goal $g$. To define our abstract MDPs, we employ hindsight labeling and state abstraction with LLM-based operators.

### 3.1.1 STATE ABSTRACTION

The goal of state abstraction is to reduce the size of the environmental state space by grouping together similar states in a way that reduces the complexity of the underlying problem being solved (Li et al., 2006). With a well-designed state abstraction, RL algorithms can learn more efficiently, requiring fewer samples of data, which is particularly relevant in our offline setup. Formally, let $F : \mathcal{X} \times \mathcal{G} \to \mathcal{S}^g$ be an abstraction operator that given a goal $g$ maps a single observation $x \in \mathcal{X}$ to an abstract state $s^g = F(x; g)$. The abstraction map, $F$, should be such that $|\mathcal{X}| \gg |\mathcal{S}^g|$.

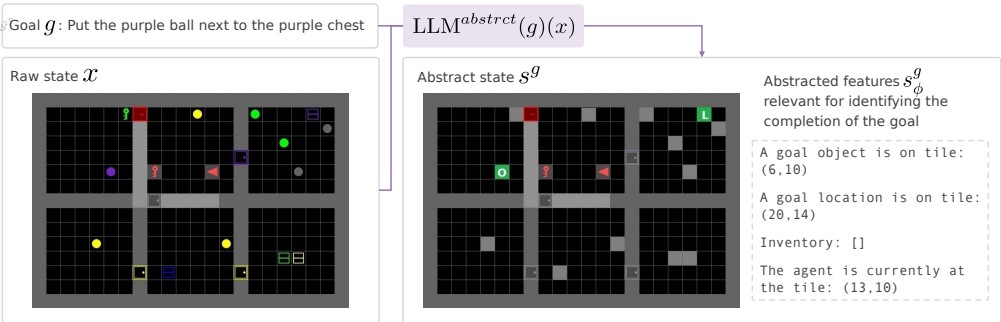

Figure 2: *An example of state abstraction for a grid world.* The LLM-induced abstraction function reduces the complexity of the original state by treating irrelevant distractors as walls, disregarding the color of opened doors, and identifying the object to be picked up (marked with "O") and its designated location (marked with "L").

In this paper, we use natural language to guide state abstraction, so that the resulting abstract states contain only the goal-relevant information. Namely, we consider environments that can be represented in a $d$-dimensional feature space, $\mathcal{X} = \mathcal{X}^1 \times \mathcal{X}^2 \times \ldots \times \mathcal{X}^d$. We assume that only a relatively small subset of these variables is relevant for solving a specific goal $g$ and an even smaller subset is required to identify the goal states $\phi(g)$.

The state abstraction operator is implemented as a collection of Python functions built on top of the feature selection made by a prompted language model, $F(\,\cdot\,; g) = \text{LLM}^{abstrct}(g)(\cdot)$. Using LLM powered Python functions instead of directly applying the LLM to create an abstraction of each state reduces the number of LLM calls from $|\mathcal{X}||G^{tr}|$ to $|G^{tr}|$ and ensures that the abstraction is consistent across all states. The prompt for generating the code includes contextual information about the environment, a list of features, and a description of the goal, instructing the LLM to create a function that removes features of a state that are irrelevant to achieving the specified goal. Additionally, state abstraction functions separate the set of relevant features into two subsets: $s^g_\phi$ and $s^g_{\bar{\phi}}$, so that $s^g_\phi \cup s^g_{\bar{\phi}}$. The features in $s^g_\phi$ are the ones which are necessary for identifying if the underlying low-level state achieves $g$, i.e. if $x \in \phi(g)$. The remaining features, which are relevant for solving the task specified by $g$ but not strictly necessary for identifying if this goal is achieved are found in $s^g_{\bar{\phi}}$. We introduce this separation of abstracted features to even further reduce the dimensionality of the state space for hindsight labeling (see next section). Figure 2 shows an example effect of applying our LLM state abstraction on a grid-world from the BabyAI environment.

### 3.1.2 GOAL-CONDITIONED HINDSIGHT LABELING.

Following recent works (Kwon et al., 2023), we hypothesize that the existing abilities of LLMs in natural language understanding are sufficient to perform the simple task of identifying whether a particular state belongs to the set of goal states $\phi(g)$ associated with a given goal description $g$.

In order to perform hindsight labeling of our dataset $\mathcal{D}$, we wish to approximate $\phi$, and thus the reward $R_\phi(\cdot; g)$, with a prompted language model $\text{LLM}^{rwrd} \approx \mathbb{1}_{(g \text{ is achieved in } s^g)}$. Note, we assign the rewards in the abstracted spaces $\mathcal{S}^g$ and not the original state space $\mathcal{X}$. Given the large number of goals and states, to reduce the number of LLM calls needed, instead of using language models directly, we train proxy reward models–lightweight neural networks trained to predict the labels generated by the prompted language model, $\text{LLM}^{rwrd}$. For each goal $g$, we build a supervised dataset, $\{(s^g, r^g) : r^g = \text{LLM}^{rwrd}(s^g; g), s^g \in \tilde{\mathcal{S}}^g\}$, where $\tilde{\mathcal{S}}^g$ is a small, diverse subset of the abstract space $\mathcal{S}^g$ and $r^g \in \{0, 1\}$. We train a neural network $R_\theta(\cdot; g) : \mathcal{S}^g \to \{0, 1\}$ to predict binary rewards for the entire abstract state space $\mathcal{S}^g$. The subset $\tilde{\mathcal{S}}^g$ is chosen so that for any two abstract states, the set of features relevant for goal-identifications is distinct, i.e. $\forall s_1^g \neq s_2^g \in \tilde{\mathcal{S}}^g, s_{1,\phi}^g \neq s_{2,\phi}^g$. This maximizes the chances of including goal-states in $\tilde{\mathcal{S}}^g$, mitigating the potential issue of generating a highly-imbalanced dataset for training our proxy neural networks. These proxy reward functions provide a much more cost-effective way to perform hindsight labeling compared to labeling all states from $\mathcal{D}$ for all goals from $\mathcal{G}^{tr}$ directly by LLM prompting or with human annotators. Appendix D shows that for the BabyAI environment, depending on the goal, proxy rewards reach near 100% accuracy in comparison to the ground truth rewards of the environment.

## 3.2 STEP 2. SOLVING THE ABSTRACT MDPS

After applying state abstractions and hindsight labeling to our offline dataset $\mathcal{D}$, for each goal $g \in \mathcal{G}^{tr}$, we obtain an offline dataset $\mathcal{D}^g := \{(s^g, a, s^g, r^g)\}$. Given these data, we can apply any offline reinforcement learning method to learn optimal policies $\pi^g$, for each goal $g \in \mathcal{G}^{tr}$. In practice, however, to learn the goal-conditioned policies, the chosen RL method should be scalable, as we must solve multiple MDPs, one for each goal in $\mathcal{G}^{tr}$. Therefore, in our instantiation, we discard computationally intensive methods. Furthermore, as the generalization to new states is tackled by the next step, we do not require at this stage that the learned policies generalize to unseen states. Given these considerations, we simply choose tabular Q-learning (Watkins & Dayan, 1992) to solve the set of abstract MDPs. At the end of this stage, we obtain a set of learned policies $\{\pi^g : g \in \mathcal{G}^{tr}\}$. These policies are limited to the set of training goals and the set of states observed in $\mathcal{D}$. The final step of our pipeline addresses these limitations.

## 3.3 STEP 3. TRAINING THE LLM AS A GOAL-CONDITIONED POLICY

To enable generalization to previously unseen states, and more importantly, generalization to novel goals, the final step of our method distills the knowledge of the optimal actions per each abstract state and goal into a pre-trained LLM. We build a supervised dataset $\mathcal{D}^{SFT}$ consisting of goal commands, initial abstract states and the sequence of optimal actions with respect to the learned policies. Concretely, we have

$$\mathcal{D}^{SFT} := \{(g, s_0^g, [a_0^{*,g}, \ldots, a_{n_g}^{*,g}]) : \quad g \in \mathcal{G}^{tr}, \ s_0^g \in \mathcal{D}^g, \ a_t^{*,g} = \arg\max_{a \in \mathcal{A}} \pi^g(a \mid s_t^g),$$

$$s_{t+1}^g = \arg\max_{s \in \mathcal{S}^g} \hat{P}^g(s|s_t^g, a_t^{*,g})\}, \ n_g \ s.t. \ R_{\hat{\theta}}(s_{n_g+1}^g; g) = 1\},$$

where $\hat{P}^g$ is the empirical state transition function based on the abstract datasets $\mathcal{D}^g$, obtained during Q-learning in step 2. We then fine-tune a pre-trained large language model on $\mathcal{D}^{SFT}$ using the standard next-word prediction objective. We integrate description of the goal $g$ and the state $s_0^g$ into a prompt and set the sequence $[a_0^{*,g}, \ldots, a_{n_g}^{*,g}]$ as the expected completion. We expect that the fine-tuned language model combined with the state abstraction function $\text{LLM}^{abstrct}$ can effectively act as a proxy for the general, goal-conditioned policy $\pi^*$ from equation (2), generalizing to any new goal $g \notin \mathcal{G}^{tr}$ and previously unobserved low-level state $x \in \mathcal{X}$.

## 4 RELATED WORK

**LLMs for decision making.** There is growing interest in using general-purpose LLMs directly as decision-making agents (Yao et al., 2023). Various prompting techniques, such as chain of thought (Wei et al., 2023) and self-reflection (Ji et al., 2023), have been developed to enhance LLMs' abilities in long-term planning tasks. However, as demonstrated in previous works (Szot et al., 2024; Finn,

2024), prompting techniques alone are insufficient for solving complex decision-making tasks in dynamic environments. To effectively utilize the knowledge embedded in LLMs for RL problems, these models must be grounded in the dynamics of the environment. This grounding can be achieved either through in-context learning (Wang et al., 2023; Wu et al., 2023) or fine-tuning (Carta et al., 2023; Tan et al., 2024; Brohan et al., 2023a). A key limitation of in-context learning is its restricted window size. In this work, we focus on fine-tuning; however, unlike prior studies, we significantly reduce the requirements on input data for fine-tuning the decision-making agent.

**LLMs as data enhancers.** To apply conventional RL methods in search of optimal goal-conditioned policies, we must augment our dataset of state-action transitions with goal-dependent rewards. This process, known as hindsight labeling, has traditionally been performed manually by human annotators or through learning the reward function from expert demonstrations (Ziebart et al., 2008; Fu et al., 2018; Bahdanau et al., 2018). Recent studies, however, have demonstrated that task-specific rewards can be effectively generated using pre-trained LLMs (Yu et al., 2023b; Ma et al., 2023; Xie et al., 2023). While successful, most LLM-based approaches rely on iterative prompting strategies, which are costly in terms of LLM calls. Our approach to hindsight labeling reduces this cost by approximating the LLM-induced reward function with a lightweight neural network. Furthermore, we assign rewards in abstracted state spaces, significantly reducing the number of states to be labeled. Similar to the work of Peng et al. (2023), our state abstraction function uses the language command to guide the elimination of irrelevant state features.

**Language-conditioned RL.** Numerous previous studies have explored learning language-conditioned policies by assuming access to ground-truth environment rewards (Jiang et al., 2019; Co-Reyes et al., 2018), real-time experimentation (Fu et al., 2018; Bahdanau et al., 2018; Mirchandani et al., 2021), or expert demonstrations paired with language annotations (Stepputtis et al., 2020; Lynch & Sermanet, 2021; Xiao et al., 2023; Brohan et al., 2023b;a). In contrast, our approach aims to learn language-conditioned policies from entirely offline datasets, which may be highly suboptimal and which are unlabeled, with no environment- or human-provided reward signals. Regarding policy evaluation, much of the prior work in language-conditioned RL and IL tests agents on new language commands synonymous with those seen during training (Lynch & Sermanet, 2021; Nair et al., 2022). Similar to the works of (Xiao et al., 2023; Brohan et al., 2023a; Shridhar et al., 2021a; Jang et al., 2022), our focus lies on novel instructions corresponding to previously unsolved goals.

Refer to Appendix A for an extended discussion of the related work.

## 5 EXPERIMENTS

**Questions.** In our experiments we aim to answer the following questions: **(Q1)** Does the use of a pre-trained language model enable generalization to new language commands and new states? **(Q2)** How does our method compare to simpler prompting-based methods and alternative approaches to language-conditioned RL? **(Q3)** As a result of SFT, does the language model memorize the optimal actions or does it learn *generalizable* and *compositional* skills? **(Q4)** How does our method scale with computer power and what is the effect of language abstractions on data efficiency?

**Experimental Setup.** For the experiments we require a controlled environment where a wide variety of distinct goals can be specified, and where the semantics of the states and actions are easily interpretable by an LLM. Thus, we choose the BabyAI environment (Chevalier-Boisvert et al., 2018), a grid world platform for instruction following where an agent receives natural language goal instructions such as: *"Go to the tile (3,2)"*, *"Pick up the blue key"* or *"Look behind the green locked door"*. A detailed discussion on the environment choice can be found in Appendix A. The grids can consist of multiple rooms connected by open or locked doors and different distractor objects that the agent can interact with (boxes, keys, balls, etc.). The action space $\mathcal{A}$ consists of several navigation primitives (`forward`, `pickup`, etc.). In our setup, we assume full observability of the original state space. Each state can be represented as a long list of features and their coordinates (see Appendix B.2 for example state representations in a text format).

**Metrics.** We rely on the following metrics to evaluate our learned policies: *success rate*: proportion of attempts in which the agent achieves the goal within the time limit (500 steps); *episode Length*: the average number of steps taken to reach the goal or the time limit; *invalid actions*: ratio of invalid actions (e.g., moving into a wall) to total actions.

## 5.1 Q1: ONLINE EVALUATION: GENERALIZATION BENCHMARK

**Setup.** We choose the collection of `Synth` environments from BabyAI as the main test bed for TEDUO. All environments are constructed as a 22x22 grid and containing 9 rooms. They differ in the type, position, and color of the distractors. The tasks include goals such as "go to the {color} {object}", "pick up the {color} {object}", or "put the {color} {object} next to the {color} {object}". We use a list of 500 goals as $\mathcal{G}^{tr}$. The set of testing goals contains 100 goals that are semantically distinct from those in $\mathcal{G}^{tr}$. We also augment the set of testing goals by asking GPT-4 to paraphrase the original commands provided by BabyAI. We train a Llama-3-8B model with TEDUO based on a dataset $\mathcal{D}$ containing 800k non-unique state-action-state triplets generated according to a policy that is a random mixture of default policies from BabyAI (see Appendix B.1 for details).

**Baselines.** We compare our fine-tuned Llama-3-8B agent with non-fine-tuned LLMs: Llama-3-8B and Llama-3-70B using a) vanilla and b) chain-of-thought prompting Wei et al. (2023) with additional demonstrations provided in-context (in-context+CoT). The latter integrates expert demonstrations generated during step 2 of TEDUO to test the in-context learning ability of the LLM. Following recent works Mezghani et al. (2023); Li et al. (2022); Cao et al. (2023), we also compare against BabyAI-IL-bot, the baseline proposed by the authors of BabyAI (Chevalier-Boisvert et al., 2018), which is the combination of a GRU to encode the instruction, CNN+FILM layers to encode the grid and an LSTM memory. We train this method via imitation learning on the policy generated by TEDUO, steps 1&2. Implementation details of the baselines can be found in Appendix B.6.

Table 1: *Online evaluation of generalization performance.* Results averaged over 400 $(g, s_0^g)$ pairs. Standard error in brackets.

| Method | Environment | Goals | Success Rate [%] | Episode Length | Invalid Actions [%] |
|---|---|---|---|---|---|
| Llama-3-8B (vanilla) | train/test | train/test | 17 (±0.9) | 444 (±3.2) | 42 (±0.1) |
| Llama-3-70B (vanilla) | train/test | train/test | 14 (±0.7) | 452 (±3.0) | 55 (±0.2) |
| Llama-3-8B (in-context+CoT) | train/test | train/test | 16 (±0.7) | 443 (±3.3) | 42 (±0.1) |
| Llama-3-70B (in-context+CoT) | train/test | train/test | 21 (±0.9) | 432 (±3.8) | 47 (±0.3) |
| TEDUO: steps 1 & 2 + BabyAI-IL-bot | train | train | 69 (±1.2) | 248 (±4.9) | 17 (±0.6) |
| | test | train | 45 (±1.2) | 344 (±4.8) | 19 (±0.6) |
| | train | test | 15 (±0.8) | 453 (±2.9) | 44 (±0.7) |
| | test | test | 16 (±0.8) | 447 (±3.1) | 36 (±0.6) |
| **TEDUO**-Llama-3-8B | train | train | 65 (±1.4) | 203 (±6.7) | 21 (±0.7) |
| | test | train | 53 (±1.1) | 257 (±5.4) | 27 (±0.7) |
| | train | test | 55 (±1.6) | 241 (±7.5) | 22 (±1.1) |
| | test | test | 45 (±1.3) | 286 (±6.1) | 31 (±1.2) |

**Results.** Based on the results presented in Table 1 we make the following observations:

- **Prior knowledge of LLMs is insufficient.** We find that non-fine-tuned LLMs, irrespective of their parameter count or prompting method struggle in solving tasks from the BabyAI environments. Low success rate and high invalid action ratios indicate the inability of LLMs to understand the dynamics of the environment. Common failures include only using the action "move forward" without considering the agent's direction or attempting final actions (e.g. door opening) without first navigating to the correct location, The observed poor performance, underscores the need for developing data-efficient methods of distilling knowledge of the environment-dynamics into LLMs. Our fine-tunning strategy brings the success rate of the Llama-3-8B language model from 17% to 65% in its training setting and 45% for the in-the-wild setting.

- **Generalization to new environments and goals.** We further look at the generalization abilities of our fine-tuned TEDUO-Llama-3-8B model to new environments and goals. When the LLM is tested on new environments unseen during training, a performance drop of 12% is observed, significantly lower than BabyAI-IL-bot baseline with a drop of 24%. This difference can be explained by the RL baseline's overfitting due to the limited offline training data, while TEDUO-Llama-3-8B benefits from the zero-shot capabilities of the pretrained LLM. This effect is even more pronounced with new goals, where TEDUO experiences only an 8% decrease in success rate, compared to a 40% drop for the BabyAI-IL-bot. Overall, TEDUO achieves nearly three times better performance than the RL baseline when generalizing to both new natural language commands and environments. We analyze success rates per goal type in Appendix D.1.

## 5.2 **Q2**: ONLINE EVALUATION: ABLATION STUDY

**Setup.** Using the same experimental setup as in the previous section, we compare our full fine-tuning pipeline with its ablations. After obtaining the abstract datasets $\mathcal{D}^g$ with the first step of TEDUO, we generate goal-conditioned policies with naive behavioral cloning (step 1 + GCBC). We also compare our fine-tuned Llama-3-8B against the performance of the GCRL policies obtained with offline Q-learning in step 2. Note, neither of GCBC nor GCRL can generalize to new, previously unseen language commands. Therefore, in this study, we are only looking at performance on goals from $\mathcal{G}^{tr}$. Ablation of the abstraction function is delayed to the next section.

Table 2: *Ablation study.* Results averaged over 400 $(g, s_0^g)$ pairs. Standard error in brackets.

| Method | Success Rate [%] | Episode Length | Invalid Actions [%] |
|---|---|---|---|
| Step 1 + GCBC | 7 (±0.6) | 474 (±2.3) | 11 (±0.1) |
| Steps 1 & 2 (GCRL) | 16 (±0.8) | 430 (±3.9) | 10 (±0.1) |
| All steps Llama-3-8B | 65 (±1.4) | 203 (±6.7) | 21 (±0.7) |

**Results.** The results of GCBC and GCRL can be seen as ablations of our pipeline. We first note that the success rate of naive behavioural cloning is low, indicating low fidelity of the data collection policy and highlighting the need for incorporating offline policy-learning methods. Moreover, the significantly improved performance of the Q-learning policies (GCRL) validates the effectiveness of the first two steps within our pipeline. Thus, the synthetically constructed abstract MDPs are meaningful offline constructs that yield, given the data available, optimal policies effective during online testing. Finally, the improved performance of our fine-tuned Llama-3-8B over the Q-learning/GCRL policies on training goals and environments confirms the importance of the third step of our method and suggests that the ungrounded, prior knowledge of large language models improves generalization to new previously unseen states.

## 5.3 **Q3**: LEARNING AND EXPLOITING CORE SKILLS

The aim of the first part of our experiments is to investigate the generalization abilities of the LLM fine-tuned with our pipeline. We wish to investigate if by learning the optimal policies for diverse goals and environments, the LLM can integrate the core skills required to achieve these goals and how such skills can be transferred across tasks. We also investigate the aspect of skill compositionality. Does the prior knowledge of the LLM, now grounded in the environment dynamics, suffice to compose together learned skills to solve novel tasks?

### 5.3.1 SKILL TRANSFER AND COMPOSITIONALITY.

**Setup.** We are working with the following three types of illustrative environments:

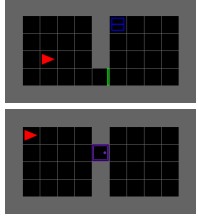

**Type A**: A grid with 2 rooms, an open door and a box. The language commands are of two types: "go to the tile (x,y)" and "pick up the {color} box".

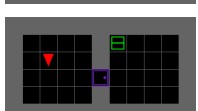

**Type B**: A grid with 2 rooms and a closed door. The language commands are the types: "go to the tile (x,y)" and "open the {color} door". The agent and the object are in different rooms, so the agent must pick up the key and open the door first.

**Type C**: A grid with 2 rooms, a closed door and a box. The language commands are the types: "pick up the {color} box". The agent and the object are in different rooms, so the agent must open the door first and then pick up the box.

The position and color of the door and box vary across different instantiations of the environments. We use type A and B environments for training and type C environments for testing. We note that tasks from type C environments require the internalization of three core skills: moving to a given location, opening a door, picking up a box. The skill of moving to a location can be obtained from both environments A and B, but the skill of picking up a box or opening the door can only be obtained from one of the environments, A or B, respectively. This setup allows us to investigate the transferability of learned skills across environments and their compositionality.

Table 3: *Performance on test tasks from type C environments.* TEDUO A and TEDUO B have been trained in only one environment whereas TEDUO A&B has been trained in both.

| Method | Success Rate [%] | Episode Length | Invalid Actions [%] |
|---|---|---|---|
| LLM (vanilla) | 0 (±0.0) | 20 (±0.0) | 75 (±0.5) |
| TEDUO A | 0 (±0.0) | 20 (±0.0) | 51 (±0.7) |
| TEDUO B | 0 (±0.0) | 20 (±0.0) | 28 (±0.4) |
| TEDUO A&B | 60 (±2.1) | 16 (±0.2) | 37 (±1.0) |

**Results.** Table 3 demonstrates that the LLM fine-tuned on tasks from both Type A and B environments achieves a 60% success rate, compared to the non-fine-tuned baseline, which fails entirely. Although the environment is simpler, baseline performance is lower than in Table 1 due to shorter maximum episode length (20 vs. 500). The agent trained only in Environment A (TEDUO A) reaches a 99% success rate in tasks without closed doors (Type A grids), but consistently fails when the goal is

behind a door. Similarly, TEDUO B achieves an 81% success rate in new grids from Type B but cannot generalize to Type C. The observation that TEDUO A&B can generalize to a new environment C that requires a combination of both skills independently seen during training indicates that the fine-tuned LLM does not merely memorize optimal trajectories for individual tasks. Instead, it learns core, generalizable abilities that can be combined to address novel tasks in new settings. This result contrasts with the failure of an LLM trained in only one environment, emphasizing the significance of multi-skill learning for successful generalization, which our framework enables.

### 5.3.2 INTERNALIZATION OF CORE SKILLS.

One of the core skills required to successfully solve tasks from the BabyAI environments is to identify whether the agent at its given location is facing an object or a wall, or it is free to move forwards. This section provides additional insights into the behaviour and internal representation of states of the LLM fine-tuned with TEDUO in comparison to a base LLM.

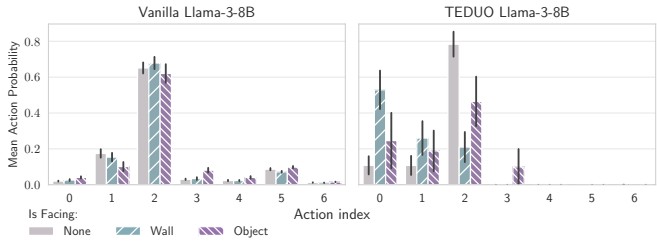 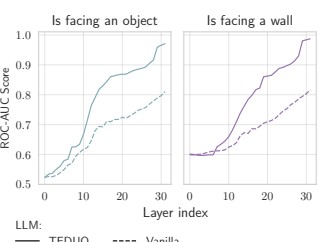

(a) *Action probabilities.* The action codes are: 0: turn left, 1: turn right, 2: move forward, 3: pick up an object, 4: drop an object, 5: toggle an object, 6: done completing the task

(b) *ROC-AUC score of the linear probe.*

Figure 3: *Interpretability results for detection of walls and objects.*

**Setup.** We operate within the `Synth` BabyAI environments as in the main evaluation benchmark and generate a dataset consisting of 10 random goals and 512 states per each goal. We embed each goal-state pair into our prompt template for eliciting actions and fine-tuning the language models and pass them through both the base and fine-tuned Llama-3-8B from experiments 5.1 and 5.2. We record the logprobabilities of the tokens $[0, 1, \ldots, 6]$ as well as the hidden representation of states at each layer. We label our dataset according to whether at the given state the agent is facing a wall, an object, or it is free to move forwards. For each layer, we fit two linear probes on top of the hidden representations: one to predict if the agent is facing a wall and the other if it is facing an object.

**Results.** First, from Figure 3(a), we observe that a non-fine-tuned Llama-3-8B puts a high probability on the action 'move forwards' irrespective of whether the agent is facing an obstacle or not; this results in a high ratio of invalid actions, as previously observed in the benchmark experiments. After fine-tuning with TEDUO, the probability of moving forwards when facing an obstacle is significantly reduced, putting more weight on the actions of moving left or right to avoid the obstacle. We also observe, that our TEDUO method taught the LLM that objects can be picked-up (action 3), only when the agent is directly facing it. From the linear probe experiments (Figure 3(b)) we observe that after fine-tunning, the internal representations of states directly encode the information of whether the agent is facing an obstacle. At the final layers, the ROC-AUC score of predicting both types of labels is near 100%, in sharp contrast with the score of around 80% for the non-fine-tuned model. Yet, the score of 80% is still relatively, high, indicating that the original state representations are sufficient to identify whether the agents is facing an obstacle, but, since the non-fine-tuned

LLM lacks grounding of this knowledge with respect to the environment dynamics, it struggles to translate it into an optimal action to be taken. This result underscores the claims of previous works that out-of-the-box LLMs struggle to translate their prior knowledge into low-level actions within dynamic environments (Finn, 2024; Szot et al., 2024).

## 5.4 **Q4**: Data efficiency and Scaling of TEDUO

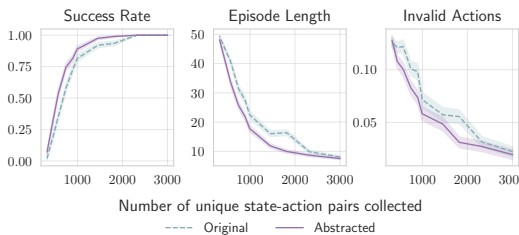

Figure 4: *Performance vs. offline dataset size. The abstraction function enhances data efficiency.*

Table 4: *Performance vs. compute power / number of training goals.*

| TFlops | $|\mathcal{G}^{tr}|$ | Success Rate [%] | Episode Length | Invalid Actions [%] |
|---|---|---|---|---|
| 5.2e7 | 266 | 33 | 342 | 32 |
| 8.6e7 | 372 | 36 | 330 | 40 |
| 1.4e8 | 534 | 45 | 286 | 31 |

**Impact of state abstraction.** In Figure 4, we look at the performance of the learned Q-learning policies (i.e. the policies $\{\pi_g\}_{g \in \mathcal{G}^{tr}}$ obtained at the end of TEDUO step 1+2 for different sizes of observational dataset $\mathcal{D}$. This experiment is realized with and without the ablation of the abstraction function during step 1. As anticipated, the efficacy of the learned policies improves with the increasing size of the dataset $\mathcal{D}$ until reaching a plateau. On average, our LLM-based state abstraction function reduces the number of unique states within each abstract MDP by 10% (Fig. C.8 in the Appendix). Due to the reduction in state space size, the abstraction function significantly enhances the data efficiency of our training method across all three performance metrics. Furthermore, the size of the state spaces $\mathcal{S}_\phi^g$, corresponding to the subset of features relevant for identifying the completion of the goal is reduced to just around 20% of the original state space size (Fig. C.8 in the Appendix). This reduces the size of the goal detection datasets for training and the subsequent goal-identification described in section 3.1.2 5-fold.

**Compute power is the new bottleneck.** Given a fixed observational dataset $\mathcal{D}$, we can expand at no extra cost the fine-tuning dataset $\mathcal{D}^{SFT}$ by introducing more training goals in $\mathcal{G}^{tr}$. Yet, larger $\mathcal{D}^{SFT}$ necessitates more compute power for training the LLM agent. Table 4 demonstrates the scaling of our method with compute power. As expected, training on a wider range of goals results in an improved performance on unseen test goals. We do not observe a plateau in performance metrics, suggesting that with additional compute further gains may be possible. Consequently, our approach shifts the bottleneck from the limited availability of real observational data to computational power.

## 6 Discussion

**Limitations.** Leveraging LLMs' prior knowledge enables efficient policy generation with minimal data. However, some applications may benefit more than others. First, certain scenarios may be out of distribution even for LLMs trained on extensive Internet data. Second, we assume that the environment state can be represented textually, which, although feasible for many applications due to language's expressiveness, may not be ideal in all cases. Third, due to the discrete nature of LLM tokenization, using fine-tuned LLMs to directly output actions requires discretization of the action space, which can hinder performance in continuous control tasks. Lastly, while data requirements are minimal, they still assume some practitioner knowledge of the environment and the data $\mathcal{D}$ to propose training goals $\mathcal{G}^{tr}$ likely achievable in $\mathcal{D}$ (see Appendix C.1 for details).

**Conclusions.** TEDUO introduces a novel framework for developing natural language instruction-following agents capable of generalizing to new states and instructions in a zero-shot setting, using a collection of unlabeled state-action transitions. This is the first RL pipeline to create natural language goal-conditioned policies in an offline setting using unlabeled data. It surpasses the current state-of-the-art in zero-shot generalization for both goal and domain adaptation. To achieve this, we propose a data-driven approach to teach environment dynamics to a large language model, enabling it to interact effectively with the environment—a task where LLMs typically underperform. This result could potentially extend the applicability of LLMs to new domains requiring multi-step reasoning and dynamic interaction with the environment.

**Reproducibility statement.** The environment used to analyze and benchmark the methods are publicly available. Within the TEDUO pipeline and for benchmarking we use open-source language models. Every step of our method is clearly stated in Section 3. Additional details to reproduce the experiments, including prompts and implementation of baselines, are presented in Appendix B. The code for this paper is provided as supplementary material.

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

## A  EXTENDED RELATED WORK

### A.1  GENERALIZATION IN OFFLINE REINFORCEMENT LEARNING

Following the work of Mediratta et al. (2024), we separate the generalization abilities of offline reinforcement learning algorithms into two categories: new instruction following and adaptation to new states or environments.

**Goal-conditioned RL.** Goal-conditioned Reinforcement Learning (GCRL) is a subfield of RL dedicated to developing policies capable of achieving multiple goals within the same environment dynamics. These policies are conditioned on an additional input, $g$, indicating the goal that the next action should aim to achieve. While most recent research has focused on online settings (Islam et al., 2022; Han et al., 2021; Hong et al., 2023; Yang et al., 2021), only a few methods have addressed the offline GCRL problem (Yang et al., 2022; Ma et al., 2022; Chebotar et al., 2021). (Yang et al., 2023) offers a comparison of these methods and highlights the key challenges in offline GCRL. Additionally, these approaches typically restrict goal representations to those expressible as a single state in the state space (Chebotar et al., 2021), a scalar parameter (Ma et al., 2022), or a fixed set of known goals (Yang et al., 2022).

**Language-conditioned RL.** Our work addresses the problem of goal-conditioned RL, where goals are expressed in natural language. While using language to specify goals is natural and broadens the range of possible goals it comes with the challenge of grounding the semantics of language in the environment state space and dynamics. Such language-instruction following agents have been widely studied in both reinforcement learning and imitation learning contexts. However, most existing methods either rely on access to an online environment for interaction (Fu et al., 2018; Bahdanau et al., 2018; Jiang et al., 2019; Mirchandani et al., 2021) or require costly, goal-annotated expert datasets of offline demonstrations (Stepputtis et al., 2020; Lynch & Sermanet, 2021; Xiao et al., 2023; Brohan et al., 2023b;a). In contrast, our approach does not assume any environment-provided reward signal or access to real-time exploration. Furthermore, in terms of generalization to new natural language instructions, we distinguish between evaluation on instructions that simply paraphrase the training goals (Nair et al., 2022; Lynch & Sermanet, 2021) from those that represent semantically novel goals. Similar to the works of Xiao et al. (2023); Brohan et al. (2023a); Stepputtis et al. (2020); Shridhar et al. (2021a); Jang et al. (2022), our focus is on the latter, more challenging scenario.

**Domain Generalization.** While the previous section addressed generalization to new goals, this section focuses on generalization to novel state-action transitions. This type of generalization extends beyond goal-conditioned RL, as it is essential even for single-goal RL. It has been widely studied and observed that Offline RL methods often overfit to the training distribution of state-action transitions, resulting in poor performance when the test distribution differs. Various approaches have been proposed to address this distribution shift, including regularization techniques Kostrikov et al. (2021); Kumar et al. (2020), model-based RL Yu et al. (2020); Kidambi et al. (2021), and enhanced representation learning Mazoure et al. (2021); Fan & Li (2022). In TEDUO, we intentionally avoid domain generalization when solving the abstract MDPs in step 2 to prevent providing incorrect examples to the LLM in step 3. However, our method achieves domain generalization by leveraging the zero-shot capabilities of the fine-tuned LLM. Future work could enhance TEDUO by replacing tabular Q-learning in step 2 with a method that generalizes to new state-action transitions.

### A.2  OFFLINE POLICY LEARNING WITH MINIMAL DATA REQUIREMENTS.

This paper focuses on realistic requirements regarding the training inputs. We work under offline setting, with a limited number of unlabeled environment transitions (i.e., $(x_t, a_t, x_{t+1})$ triplets) and without any assumptions about the policy that generated the actions. To address this scenario, we employ LLMs to label the data, enabling the use of RL methods, and as an abstraction function to enhance sample efficiency. Below we discuss the related work regarding these two steps.

**Hindsight labeling**. Labeling data for goal-conditioned RL requires the design of reward functions for each goal. The most common approach for designing the rewards relies on handcrafted methods that are often require multiple refinements through trial and error (Knox et al., 2022). With a large number of goals, manual reward design becomes infeasible. Inverse Reinforcement Learning (Ziebart et al., 2008; Fu et al., 2018) attempts to generate reward functions directly from data, but it

requires a large amount of expert demonstrations. Recent studies have explored the use of LLMs and VLMs as reward functions. These methods typically involve creating a preference dataset (Klissarov et al., 2023), comparing the cosine similarity between natural language goals and state representations, or leveraging the coding abilities of LLMs (Yu et al., 2023a), especially in an online iterative fashion (Ma et al., 2023; Xie et al., 2024). These approaches are however aimed at densifying the reward signal. In contrast, our method requires generating reward labels for a large number of goals (approx. 100-1000), making the scalability of the process crucial. Therefore, we focus on generating rewards with a limited number of LLM calls. Our approach relies on LLM-based detection of task completion, which has been proven effective by Kwon et al. (2023). We further reduce the number of LLM calls by approximating the LLM-generated rewards with lightweight proxy neural networks.

**State abstraction.** State abstraction aims to reduce the complexity of the state space by eliminating irrelevant features, thereby improving the efficiency of learning algorithms. Early work in this area focused on state aggregation, where similar states are grouped together to form more compact representations, with state similarity defined through the transition dynamics, value- or Q-functions (Andre & Russell, 2002; Li et al., 2006; Givan et al., 2003; Abel et al., 2018). Recent advancements have explored more sophisticated methods, such as deep learning-based state abstractions, employing neural networks to learn abstract representations of states (Allen et al., 2021). In this work, we explore the use of LLMs to accomplish the task of state abstraction. Our approach relies on prompting a pre-trained LLM to remove the features of a state that are irrelevant in solving the given goal. Such LLM-based state abstraction has been previously shown effective in the context of robotics by Peng et al. (2023) who employ LLMs to translate the language command into a binary mask highlighting the location of the goal-object.

### A.3 LARGE LANGUAGE MODELS FOR DECISION MAKING

**Decision Transformers.** Pre-trained models based on the Transformer architecture have been widely used to address decision-making problems. However, this paper does not focus on Decision Transformer (DT) models (Chen et al., 2021). Although DTs have been applied in goal-conditioned RL and IL (Xu et al., 2022; Raparthy et al., 2023; Putterman et al., 2022), the joint modelling of goal, state, and action representations remains challenging and requires large labeled datasets. Instead of training a decision transformer, this papers leverages the prior knowledge accumulated in LLMs trained on Internet data to a) enable effective use of the limited offline, unlabeled data, b) enable generalization to previously unseen goals and states.

**General-purpose LLMs for decision making.** Utilizing off-the-shelf LLMs has gained significant attention due to its simplicity. In decision-making, LLMs have been used to create assistance functions within training pipelines to enrich data (Klissarov et al., 2023; Yu et al., 2023a; Ma et al., 2023; Xie et al., 2023; Laskin et al., 2022), and as high-level planners during inference to guide traditional RL policies (Shah et al., 2023; Ahn et al., 2022). Additionally, there has been growing interest in using general-purpose LLMs directly as decision-making agents (Yao et al., 2023). Improving the reasoning capabilities of LLM agents is now an active research area, focusing on methods that are independent of traditional RL. These include iterative prompting techniques such as self-reflection (Ji et al., 2023), chain of thought reasoning (Wei et al., 2023), and integration with planning algorithms like Monte Carlo Tree Search (Pouplin et al., 2024). Nevertheless, such methods have been shown inefficient in completing complex, multi-step decision-making tasks in dynamic environments (Finn, 2024; Szot et al., 2024). To effectively use the knowledge embedded in LLMs for solving RL problems, these models need to be grounded in the dynamics of the environment.

**Grounding LLMs with the environment dynamics.** An LLM agent grounded in an environment can link the semantics of both observations and possible actions to its internal representation system, enabling appropriate decision-making (Carta et al., 2023; Harnad, 1990). One approach to achieve such grounding is through **in-context learning**. For instance, Voyager (Wang et al., 2023) pushes the concept of an LLM agent to its limits by developing an automatic curriculum for GPT-4, supported by a library of executable programs, to play Minecraft. Another method involves providing the LLM with a game manual (Wu et al., 2023). However, these approaches either rely on extensive expert knowledge, such as carefully designed prompts, or on game manuals, which may not always be available. Additionally, in-context learning has limitations in data-driven scenarios, partly due to the restricted context window size, which is insufficient for incorporating entire datasets. An alternative

approach involves **fine-tuning** LLMs to achieve grounding. Studies such as (Tan et al., 2024) and (Carta et al., 2023) use Proximal Policy Optimization (PPO) (Schulman et al., 2017) to propose online fine-tuning of LLMs. In the robotics domain, RT2 (Brohan et al., 2023a) demonstrates that co-fine-tuning on both web-scale data and expert robot demonstrations improves performance of VLMs for decision making in the context of robotics. Our method differs from previous work by significantly lowering the requirements on input data, as we do not need online interaction or labeled expert demonstrations. Furthermore, while RT2 implements co-fine-tuning, our method utilizes an off-the-shelf pre-trained LLM, which is then fine-tuned.

### A.4 TESTING ENVIRONMENTS

This paper uses the Minigrid-BabyAI environment to benchmark its method. This choice was motivated by several factors. Most importantly, we require a sandbox environment in which a wide range of goal reaching tasks can be expressed in natural language. Robotic environments (Todorov et al. (2012); James et al. (2019)) were excluded due to precise control of robotic components being beyond LLM's prior knowledge and the need to discretize continuous actions to match LLM's tokenized output. Additionally, 3D environments (Fan et al. (2022); Puig et al. (2018)) were not considered due to computational constraints. Text-based games (Côté et al. (2018); Shridhar et al. (2021b)) were also excluded as they involve high-level text interactions, contrary to this paper's focus on low-level control task for language models.

Given the significant computational resources and time required to perform all three steps of our pipeline, in particular fine-tuning an LLM agent, our current scope is necessarily limited to BabyAI. Nonetheless, the insights derived from this controlled setting are broadly applicable and provide a foundation for future work in environments with similar tabular structures, such as NetHack (Küttler et al. (2020)) and Overcooked (Carroll et al. (2020)), which differ mainly in thematic focus (video game dungeon crawling and collaborative cooking, respectively).

## B EXPERIMENTAL DETAILS

### B.1 DATA COLLECTION

To collect the data $\mathcal{D}$ used throughout the experiments, we rely on the default goal-oriented policies from the BabyAI environment. We denote these policies by $\pi^\beta(\cdot; g)$. Our data collection policy that is random mixture of the policies $\pi^\beta(\cdot; g)$. Given a randomly sampled initial state $x_0 \in \mathcal{X}$ and an unknown goal $g$ randomly sampled from the set of original BabyAI language commands, we let the agent interact with the environment according to $\pi^\beta(\cdot; g)$ until either $g$ is reached or the limit of 500 steps is reached. This policy simulates agents attempting to accomplish multiple task within the environment. Examples of real-world unlabeled data that could be generated from such policy include CCTV footage of employees at work, logs of medical procedures performed on a patient, or YouTube videos.

Refer to Appendix C.1 for an analysis of how the data collection policy affects our pipeline, comparing goal-oriented data collection with fully random data collection.

### B.2 ABSTRACTION FUNCTION

The abstraction function utilizes contextual understanding of LLMs to identify goal-relevant features. We prompt an LLM with the given goal, a randomly sampled state representation as an exmple, and two in-context examples of the expected output. Figure B.1 shows the prompt template used. The LLM returns the goal relevant features which are then passed to a python function that processes states according to the following rules:

- Distractors identified in the selected features are labeled as either "goal object" or "goal location."

- Distractors not included in the selected features are labeled as obstacles.

- Doors not referenced in the selected features are assigned uniform colors.

- If all relevant objects are within the agent's current room, the environment outside the room is disregarded.

For this step we use the *Llama-3-70B-Instruct* language model with the following parameters: {temperature: 0, top k: 1, maximum number of tokens: 8000}.

```
<|begin_of_text|><|start_header_id|>system<|end_header_id|>You are helping a Reinforcement learning
    agent in the minigrid environment. Always answer as helpfully as possible, while being truthful
    .<|eot_id|><|start_header_id|>user<|end_header_id|>Given a grid, its features and a goal, can
    you simplify the features of the grid by detecting all the objects related to the goal and if
    necessary goal location. if necessary, make sure to flag all the relevant object and not just
    one.

I'm giving you two examples on the same grid:

Grid : "It is a 22 by 22 tiles grid. The features of the environment are:
0. The following tiles are wall: (1,7) (1,14) (2,7) (2,14) (3,7) (3,14) (4,7) (5,7) (5,14) (6,14)
    (7,1) (7,2) (7,3) (7,4) (7,5) (7,6) (7,7) (7,8) (7,9) (7,10) (7,11) (7,13) (7,14) (7,15) (7,16)
    (7,17) (7,18) (7,19) (7,20) (8,7) (8,14) (9,14) (10,7) (10,14) (11,7) (11,14) (12,7) (13,7)
    (13,14) (14,1) (14,2) (14,3) (14,4) (14,5) (14,6) (14,7) (14,9) (14,10) (14,11) (14,12) (14,13)
    (14,14) (14,16) (14,17) (14,18) (14,19) (14,20) (15,7) (15,14) (16,7) (16,14) (17,7) (17,14)
    (18,7) (18,14) (19,14) (20,7) (20,14)
1. A open purple box is on tile (1,20)
2. A open green box is on tile (5,8)
3. A open yellow box is on tile (6,5)
4. A open blue box is on tile (8,13)
5. A open purple box is on tile (15,3)
6. A open grey box is on tile (18,10)
7. A open red box is on tile (20,19)
8. A closed yellow door is on tile (4,14)
9. A closed purple door is on tile (6,7)
10. A locked grey door is on tile (7,12)
11. A closed red door is on tile (9,7)
12. A closed yellow door is on tile (12,14)
13. A closed grey door is on tile (14,8)
14. A closed grey door is on tile (14,15)
15. A closed red door is on tile (19,7)
16. A blue key is on tile (3,5)
17. A grey key is on tile (8,10)
18. A blue key is on tile (11,4)
19. A purple ball is on tile (1,16)
20. A green ball is on tile (2,20)
21. A blue ball is on tile (3,19)
22. A red ball is on tile (9,12)
23. A grey ball is on tile (9,13)
24. A yellow ball is on tile (13,1)
25. A grey ball is on tile (13,6)
26. A yellow ball is on tile (17,6)
27. Inventory : []

Exemple 1 :
The goal is "Pick up a blue key".

Following the indications, the correct output is these simplified features :

{"goal object" : (3,5) (11,4)}

Example 2 :
The goal is "Put a green box next to a grey ball".

Following the indications, the correct output is these simplified features :

{"goal object" : (18,10),
"goal location" : (9,13) (13,6),}

Now, my goal is "{goal}" and I am in the following grid :
"It is a 22 by 22 tiles grid. The features of the environment are:
{state}

Let's think step by step. First, tell me about your knowledge of the Minigrid/BabyAI reinforcement
    learning environment. Then, provide an analysis of the environment and the goal. Finally, write
    simplified features in the same format as the example.<|eot_id|><|start_header_id|>assistant <|
    end_header_id|>
```

Figure B.1: Prompt template for selecting the relevant features to achieve the goal.

```
0. The following tiles are wall: (1,7) (1,14) (2,7) (2,14) (3,7) (3,14) (4,14) (5,7) (6,7) (6,14) (7,1) (7,2)
     (7,3) (7,4) (7,5) (7,6) (7,7) (7,9) (7,10) (7,11) (7,12) (7,13) (7,14) (7,15) (7,16) (7,17) (7,18)
     (7,19) (7,20) (8,7) (8,14) (9,7) (10,7) (10,14) (11,7) (11,14) (12,7) (12,14) (13,14) (14,1) (14,2)
     (14,3) (14,4) (14,6) (14,7) (14,8) (14,9) (14,10) (14,11) (14,12) (14,14) (14,15) (14,16) (14,18)
     (14,19) (14,20) (15,14) (16,7) (16,14) (17,7) (17,14) (18,7) (18,14) (19,7) (19,14) (20,7) (20,14)
1. A open red box is on tile (1,2)
2. A open yellow box is on tile (4,9)
3. A open blue box is on tile (6,8)
4. A open grey box is on tile (16,15)
5. A open grey box is on tile (17,1)
6. A open red box is on tile (20,6)
7. A closed blue door is on tile (4,7)
8. A closed red door is on tile (5,14)
9. A closed purple door is on tile (7,8)
10. A closed blue door is on tile (9,14)
11. A closed yellow door is on tile (13,7)
12. A closed yellow door is on tile (14,5)
13. A closed red door is on tile (14,13)
14. A closed red door is on tile (14,17)
15. A closed grey door is on tile (15,7)
16. A grey key is on tile (5,20)
17. A yellow key is on tile (9,15)
18. A green key is on tile (15,5)
19. A yellow key is on tile (16,12)
20. A green key is on tile (17,15)
21. A red ball is on tile (4,19)
22. A purple ball is on tile (9,5)
23. A purple ball is on tile (12,2)
24. A blue ball is on tile (16,19)
25. Inventory : []
26. The agent is currently at the following tile: (6,10)
27. The agent is facing up
```

Figure B.2: *An example of BabyAI textualized state before state abstraction.*

```
The following tiles are wall: (1,7) (1,14) (2,7) (2,14) (3,7) (3,14) (4,14) (5,7) (6,7) (6,14) (7,1) (7,2)
     (7,3) (7,4) (7,5) (7,6) (7,7) (7,9) (7,10) (7,11) (7,12) (7,13) (7,14) (7,15) (7,16) (7,17) (7,18)
     (7,19) (7,20) (8,7) (8,14) (9,7) (10,7) (10,14) (11,7) (11,14) (12,7) (12,14) (13,14) (14,1) (14,2)
     (14,3) (14,4) (14,6) (14,7) (14,8) (14,9) (14,10) (14,11) (14,12) (14,14) (14,15) (14,16) (14,18)
     (14,19) (14,20) (15,14) (16,7) (16,14) (17,7) (17,14) (18,7) (18,14) (19,7) (19,14) (20,7) (20,14).
The following tiles are obstacles : (1,2) (4,9) (16,15) (17,1) (20,6) (5,20) (9,15) (15,5) (16,12) (17,15).
The following tiles are closed doors : (6,8) (4,7) (5,14) (6,8) (4,7) (5,14) (7,8) (9,14) (13,7) (14,5)
     (14,13) (14,17) (15,7).
A goal object is on the tile (4,19).
A goal object is on the tile (9,5).
A goal object is on the tile (12,2).
A goal object is on the tile (16,19).
Inventory : [].
The agent is currently at the tile (6,10).
The agent is facing up.
```

Figure B.3: *Textualized state from B.2 after applying state abstraction for the goal "pick up a ball".*

## B.3 REWARD SHAPING

As detailed in Section 3.1.2, the reward shaping process involves two stages.

In the first stage, a large language model LLM, *Llama-3-70B-Instruct*, is utilized to generate a supervised dataset of labeled goals. The LLM is configured with parameters {temperature: 0, top-k: 1, max tokens: 8000}, using the prompt template shown in Figure B.4. For each goal $g$, up to 5000 states are randomly sampled from $\tilde{\mathcal{S}}^g$ and labeled.

In the second stage, a collection of neural networks is trained on this dataset. The state representations are transformed from text to a grid format. The network architecture consists of a small convolutional neural network with one convolutional layer (output dimension: 32, kernel size: (2,2)), followed by two linear layers (hidden dimension: 32, output dimension: 1). A Sigmoid activation function is applied after the final linear layer, and ReLU is used after all other layers. Dropout layers are added before each linear layer. The network is trained with the following hyperparameters: learning rate of 1e-5, maximum of 3000 epochs, and dropout rate of 0.1. The dataset is split into training and validation sets (90%/10%), and the model weights with the lowest validation loss are retained.

```
<|begin_of_text|><|start_header_id|>system<|end_header_id|>You are a helpful and honest judge of good
        progress in the Minigrid/BabyAI reinforcement learning environment with respect to a specific
        GOAL. Always answer as helpfully as possible, while being truthful, simple and concise. If you
        don't know the answer to a question, don't share false information.
<|eot_id|><|start_header_id|>user<|end_header_id|>I will present you a GOAL to be achieved and the
        descriptions of a STATE of the environment. Examples of goal are "opening a door", "go to a
        specific location", "putting an object next to another other" or "picking up an object".
First, tell me about your knowledge of the Minigrid/BabyAI reinforcement learning environment related
        to the goal.
Then, write an analysis describing the semantics of the state strictly using information from the
        description and your knowledge of Minigrid/BabyAI.
Finally, respond by explicitly declaring if the state indicates that the GOAL has been achieved at
        any point in the past, writing either ("goal achieved": True), or ("goal achieved": False). If
        you have a doubt, you could also say ("goal achieved": NA).

The environment is a 22 by 22 tiles grid. An object that has been picked up is placed in the agent
        inventory.

The agent or an object is considered at an object location if it is on an adjacent tile to the object
        (for example, (4,2) and (5,3) are not adjacent as their Manhattan distance |4-5| + |2-3| = 2 is
        strictly superior to 1) or it is in the inventory. If the goal explicitly mentions the agent
        going to an object or putting an object near another object, compute the Manhattan distance,
        show the details of the computation, explicitly compare the result to 1 and then verify your
        reasoning does not have any mistakes and base your decision only on the Manhattan distance. Don'
        t say they are adjacent if their Manhattan distance is higher than 1. Don't forget to check the
        inventory. If the coordinates of the destination are mentioned, the agent must go to this exact
        tile.

For other types of goals, do not compute them and ignore the previous paragraph.

{"STATE": {state}}

{"GOAL": {goal}}<|eot_id|><|start_header_id|>assistant<|end_header_id|>
```

Figure B.4: Prompt template for labeling states as goal states or not.

## B.4 TABULAR Q-LEARNING

In TEDUO's step 2, the abstract MDPs are solved using tabular Q-learning. For each goal $g$, a Q-value table $Q^g$ of size $|S^g| \times |\mathcal{A}|$ is constructed. The Q-values are updated iteratively using the Bellman equation:

$$Q^g_{new}[s_t, a_t] \leftarrow (1 - \alpha)Q^g[s_t, a_t] + \alpha \left( R[s_t, a_t] + \gamma \max_a Q^g[s_{t+1}, a] \right).$$

The learning rate $\alpha$ is set to 0.1, and the discount factor $\gamma$ is set to 0.7. Subsequent states $s_{t+1}$ are restricted to transitions observed in $\mathcal{D}$. Iterations stop when $||Q^g_{new} - Q^g||_\infty < \epsilon$, where $\epsilon = 1 \times 10^{-6}$.

## B.5 LLM Fine-tuning

TEDUO's step 3 involves fine-tuning a large language model using the generated supervised dataset $\mathcal{D}^{SFT}$. In this paper, the fine-tuned model is *Llama-3-8B-Instruct*. We use Low-Rank Adaptation (Hu et al. (2021)) to reduce the compute cost. The hyperparameters used for the fine-tuning step are detailed in Table B.1. The model weights with the lowest validation loss are retained. The fine-tunings have been realised on a cluster of 4 A100 (80GB VRAM). The computing power provided in figure 4 is determined by multiplying the number of GPU hours by the peak Tflops (312 for A100 in bf16) and the estimated utilisation rate (90%).

Table B.1: **Fine-tuning hyperparameters**

| Hyperparameter | Value |
| --- | --- |
| Batch size (per device) | 10 |
| Learning rate | 2e-5 |
| Maximum Gradient norm | 0.3 |
| Warmup ratio | 0.01 |
| Maximum number of epochs | 3 |
| LORA rank | 512 |
| LORA alpha | 512 |
| LORA dropout | 0.1 |
| Split train/val ratio | 0.1 |
| Tensor type | bf16 |

```
<|begin_of_text|><|start_header_id|>system<|end_header_id|>You are a Reinforcement learning agent in
    the minigrid environment. You select the sequence of optimal actions to achieve the GOAL. Always
    answer as helpfully as possible, while being truthful.<|eot_id|><|start_header_id|>user<|
    end_header_id|>The state of the environment is given by the STATE. The environment is a 22 by 22
    tiles grid. The possible actions are { 0: turn left, 1: turn right, 2: move forward in the
    direction faced by the agent, 3: pick up an object, 4: drop an object, 5: toggle/activate an
    object, 6: done completing the task }.
You only output the list of numbers associated with the optimal sequence of action to achieve the
    GOAL.

STATE : {state}

GOAL : {goal}.<|eot_id|><|start_header_id|>assistant<|end_header_id|>
```

Figure B.5: This prompt template is employed to generate a sequence of optimal actions to achieve the given goal while being in the given state.

## B.6 Baselines

**BabyAI-IL-bot**. This baseline employs the official implementation from (**?**) using Imitation Learning (IL) with the largest default model parameters: memory dimension = 2028, recurrence = 80, batch size = 768, instruction architecture = AttentionGRU, instruction dimension = 256, learning rate = $5 \times 10^{-5}$. Training is performed on the supervised dataset $\mathcal{D}^{SFT}$ from TEDUO step 2 instead of an expert demonstration dataset.

**LLMs (vanilla)**. The vanilla Large Language Model baseline utilizes *Llama-3-8B-Instruct* or *Llama-3-70B-Instruct* prompted with the template shown in Figure B.5. This prompt provides basic information about the environment, current goal, and a textual (non abstracted) representation of the state.

**LLMs (in-context + CoT)**. This baseline extends the vanilla LLM approach by using the prompt in Figure B.6, which includes detailed environment information, similar to a game manual, as described in (Wu et al., 2023). It also integrates expert demonstrations using textual grid examples and goals with their optimal action sequences. The "Chain-of-Thought" (CoT) prompting technique is employed to guide the LLM through multi-step reasoning and self-reflection.

```
The state of the environment is given by the STATE. The environment is a {env[0]} by {env[1]} tiles
    grid. The possible actions are { 0: turn left, 1: turn right, 2: move forward in the direction
    faced by the agent, 3: pick up an object, 4: drop an object, 5: toggle/activate an object, 6:
    done completing the task}. An object that has been picked up is placed in the agent inventory.
    The agent or an object is considered at an object location if it is on an adjacent tile to the
    object (For example, (4,2) and (5,3) are not adjacent as their Manhattan distance |4-5| + |2-3|
    = 2 is strictly superior to 1) or it is in the inventory. If the coordinates of the destination
    are mentioned, the agent must go to this exact tile. Make sure you are facing the right
    direction before using the action "2".

You only output the list of numbers associated with the optimal sequence of action to achieve the
    GOAL.

To help you achieving the GOAL, I provide examples of optimal sequences of actions for multiple
    examples GOAL with different examples STATE.

###Example 1 :

GOAL : {Example goal 1}.

STATE : {Example state 1}.

Sequence of actions : {Example action 1}

###Example 2 :

GOAL : {Example goal 2}.

STATE : {Example state 2}.

Sequence of actions : {Example action 2}

Now, I will present you a GOAL to be achieved. First, tell me about your knowledge of the BabyAI
    reinforcement learning environment. Second, explain how you can use the proposed actions to move
    around the grid. Third, similar to the example, output a Python list that contains the sequence
    of action keys (1-6) chosen to achieve the goal.

GOAL : {goal}.

STATE : {state}.
```

Figure B.6: This prompt template is employed to generate a sequence of optimal actions to achieve the given goal while being in the given state. It uses in-context learning and Chain-of-Thought prompting.

## B.7 EVALUATION SETUP

The evaluation setup outlined in Table 1 includes training and testing configurations for both environments and goals.

**Environments.** An environment in this context refers to a grid setup, which includes the arrangement of rooms, doors, and objects. The training environments consist of the grid setups included $\mathcal{D}$. This implementation uses 40 distinct environments for training the model. Testing environments are entirely new grid setups not encountered during training. For this benchmark, we utilize 2 different grid setups for testing.

**Goals.** Training goals are defined as the goal contained in $\mathcal{G}^{tr}$, a subset of natural language instructions provided by BabyAI without any modifications. Testing goals differ both grammatically **and** semantically from training goals. They are derived from BabyAI's original instructions, distinct from $\mathcal{G}^{tr}$, and reformulated using alternative phrasings and synonyms. Tables B.2, B.3, and B.4 provide the alternative formulations and synonyms for objects and colors used in these reformulations.

Table B.2: *Alternative formulations for the natural language commands.*

| Original instruction | Alternative formulation |
| --- | --- |
| Go to the tile (X,Y) | Move to the location at the coordinate (X,Y) / Reach the position at (X,Y) / Navigate to the point (X,Y) |
| Pick up a X | Grab a X / Acquire a X / collect a X |
| Go to a X | Move to a X / Reach a X / Naviguate to a X |
| Open a X | Push a X open / Swing open a X |
| Put a X next to a Y | Set a X and a Y next to each other / Position a X alongside a Y / Place a X beside a Y |

Table B.3: *Synonyms used for the objects.*

| Original word | Synonyms |
| --- | --- |
| Box | Container / Crate / Chest |
| Key | Passcode / Lock-opener / Unlocker |
| Ball | Sphere / Globe / Orb |
| Door | Portal / Gate / Hatch |

Table B.4: *Synonyms used for the colors.*

| Original Color | Synonyms |
| --- | --- |
| Blue | Azure / Cobalt / Navy |
| Red | Scarlet / Crimson / Ruby |
| Green | Emerald / Jade / Lime |
| Yellow | Golden / Amber / Canary |
| Purple | Violet / Lavender / Mauve |
| Grey | Ash / Charcoal / Silver |

# C  ADDITIONAL RESULTS

## C.1  DATA COLLECTION POLICY

We examine the effect of the data collection policy on our pipeline's performance. Specifically, we demonstrate that our pipeline remains effective irrespective of the optimality of the data collection policy with respect to the set $\mathcal{G}^{tr}$.

Given the observational dataset $\mathcal{D}$ collected under a policy $\pi^\beta$, let $\mathcal{G}^\mathcal{D}$ represent the set of goals corresponding to goal states that have been visited in $\mathcal{D}$. This set is defined as:

$$\mathcal{G}^\mathcal{D} = \{g \in \mathcal{G} : \exists(x, a, x') \in \mathcal{D} \text{ s.t. } R_\phi(x, a, x'; g) = 1\}. \tag{3}$$

We can measure the alignment between the dataset $\mathcal{D}$ and the training goals $\mathcal{G}^{tr}$ by the size of $\mathcal{G}^\mathcal{D} \cap \mathcal{G}^{tr}$, i.e. the set of goals from $\mathcal{G}^{tr}$ that have been visited in $\mathcal{D}$. A key point is that, in step 2 of TEDUO, we cannot generate a policy $\pi^g$ for any goal $g$ not present in $\mathcal{G}^\mathcal{D}$. As discussed in section 5.4, the performance of the fine-tuned LLM depends on the size of the synthetically generated dataset $\mathcal{D}^{SFT}$, making $|\mathcal{G}^\mathcal{D} \cap \mathcal{G}^{tr}|$ an import ant metric for evaluating the fidelity of our training inputs: $\mathcal{D}$ and $\mathcal{G}^{tr}$.

To empirically analyze this, we consider two randomized policies:

- **A) Goal-oriented policy:** This is the policy used for data collection in the main experimental section. For each trajectory, a random goal from a set of goals $\mathcal{G}^\pi$ is drawn and the agents acts according to the goal-oriented policy provided in the BabyAI environment in order to achieve it. This policy simulates agents attempting to accomplish multiple task within the environment. Examples of real-world unlabeled data that could be generated from such policy include CCTV footage of employees at work, logs of medical procedures performed on a patient, or YouTube videos.

- **B) Random policy:** Actions are drawn uniformly at random from the action space. This policy represents an agent that explores the environment without a specific goal. Although this scenario is less common in real-world settings—where agents typically pursue objectives—it remains applicable forSo batch RL, particularly when learning from untrained agents with no prior knowledge.

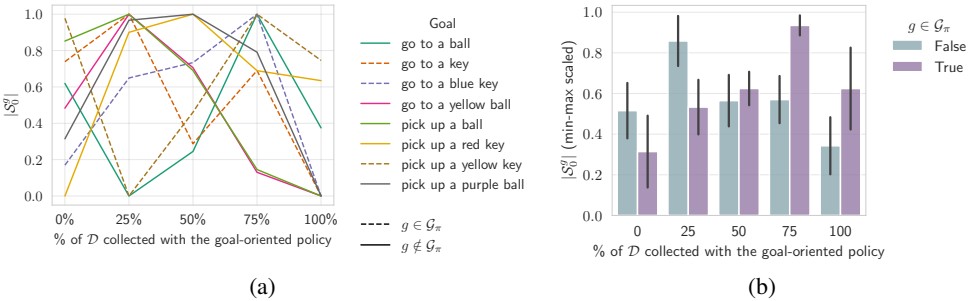

(a)  (b)

Figure C.7: *Impact of data collection policy.* The x-axis shows the proportion of data $\mathcal{D}$ collected with policy A vs. policy B for a fixed size of $\mathcal{D}$. a) The y-axis shows $|\mathcal{S}_0^g| := \{s_0^g : (g, s_0^g, [a_0^{g,*}, a_1^{g,*}, ...]) \in \mathcal{D}^{SFT}\}$, i.e. the number of unique initial abstract states $s_0^g$ for which $g$ is reachable with the learned policy $\pi^g$. Values are min-max normalized across all 5 mixture policies. b) The y-axis shows the same values as plot a), averaged across 14 goals, bars represent the standard error.

Figure C.7 illustrates that the optimal data collection policy varies by goal. For some goals policy A works better, while for others it is the fully random policy. Importantly, a comparable amount of synthetic action sequences for fine-tunning the LLM can be extracted using either policy A or B. Averaging across all goals, we find that policy A tends to perform better for goals in $\mathcal{G}_\pi$ than those not in $\mathcal{G}_\pi$. Future work could explore optimizing the set of training goals $\mathcal{G}^{tr}$ to maximize the alignment of $\mathcal{G}^{tr}$ with a given dataset $\mathcal{D}$. Yet, the necessity of aligning $\mathcal{D}$ and $\mathcal{G}^{tr}$ is moderated by

two factors. First, as shown in subsection 5.4, the abstraction function reduces the complexity of the abstract MDPs, requiring fewer data samples. Second, since extending the list of goals in $\mathcal{G}^{tr}$ is computationally inexpensive, we can continually seek better alignment.

## C.2 ABSTRACTION FUNCTION

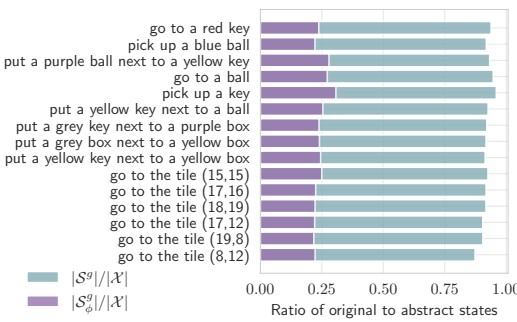

Figure C.8: Reduction in count of unique states due to applying the $\text{LLM}(g)$ abstraction functions and the relative size of the reduced abstract feature space $\mathcal{S}_\phi^g$, containing only features necessary to identify the completion of a goal $g$.

## D REWARD SHAPING EVALUATION

This section evaluates the performance of the reward-shaping step. We utilize pre-trained LLMs to identify states where a specific goal $g$ is achieved. As discussed in Section 3.1.2, the large number of states (around 200k) for each goal makes direct LLM usage impractical due to computational constraints. Therefore, the process is divided into two steps: (a) constructing a supervised dataset by labelling a subset of states (5k) using an LLM, and (b) training a lightweight neural network $R_\theta(\cdot; g)$ on this dataset.

Table D.5: *Reward Shaping Benchmark.* The accuracy, precision, and recall metrics are computed with a classification threshold ensuring at least 95% precision.

| Goals | ROC-AUG | Accuracy (%) | Precision (%) | Recall (%) |
|-------|---------|--------------|---------------|------------|
| **Go to a box** | 0.90 | 89 | 96 | 38 |
| **Pick up a ball** | 0.75 | 98 | 95 | 83 |
| **Open a door** | 0.92 | 85 | 95 | 85 |
| **Go to red door** | 0.98 | 94 | 100 | 0.2 |
| **Go to the tile (5,6)** | 1.0 | 100 | 100 | 100 |
| **Put a box next to a blue ball** | 0.64 | 100 | 100 | 25 |

Table D.5 shows the performance of $R_\theta(\cdot; g)$ for various types of goals compared to ground truth rewards. The benchmark setup is consistent with the main experiments; details are provided in the Appendix B.3. All goals achieve 95% precision, a crucial metric since false positives lead to generating incorrect data points for $\mathcal{D}^{SFT}$ in TEDUO's step 2. Conversely, false negatives only reduce data points in $\mathcal{D}^{SFT}$, which is less critical given our synthetic data abundance (see Section 5.4). Performance varies across goals; for instance, "go to the red door" has low recall (0.2%), likely due to limited positive examples in the dataset. Expanding the dataset could improve such outcomes.

## D.1 BENCHMARK RESULTS PER GOAL CATEGORY

Table D.6: *Online evaluation of generalization performance split per goal category.* This is the **success rate [%]** presented in Table 1 with the 400 $(g, s_0^g)$ grouped by goal category. Standard error in brackets.

| Method | Environment | Goals | Pick up a X | Go to the X | Open a X | Put an X next to a Y |
|---|---|---|---|---|---|---|
| Llama-3-8B (vanilla) | train/test | train/test | 11 (± 1.6) | 35 (± 2.3) | 8 (± 1.1) | 0 (± 0.0) |
| Llama-3-70B (vanilla) | train/test | train/test | 13 (± 1.7) | 33 (± 2.2) | 2 (± 0.5) | 0 (± 0.0) |
| Llama-3-8B (in-context+CoT) | train/test | train/test | 6 (± 1.3) | 36 (± 2.1) | 10 (± 1.4) | 0 (± 0.0) |
| Llama-3-70B (in-context+CoT) | train/test | train/test | 9 (± 1.4) | 45 (± 1.7) | 14 (± 1.2) | 0 (± 0.0) |
| TEDUO: steps 1 2 + BabyAI-IL-bot | train | train | 46 (± 2.0) | 92 (± 1.2) | 100 (± 0.0) | 7 (± 2.9) |
| | test | train | 30 (± 1.6) | 58 (± 1.7) | 100 (± 0.0) | 6 (± 2.1) |
| | train | test | 5 (± 1.2) | 44 (± 2.5) | 4 (± 0.9) | 0 (± 0.0) |
| | test | test | 7 (± 1.2) | 40 (± 2.2) | 5 (± 0.9) | 0 (± 0.0) |
| **TEDUO** (Llama-3-8B) | train | train | 46 (± 2.3) | 85 (± 1.3) | 100 (± 0.0) | 0 (± 0.0) |
| | test | train | 39 (± 2.4) | 65 (± 2.0) | 100 (± 0.0) | 0 (± 0.0) |
| | train | test | 20 (± 3.8) | 87 (± 3.2) | 83 (± 1.8) | 0 (± 0.0) |
| | test | test | 26 (± 3.0) | 70 (± 2.8) | 61 (± 2.6) | 0 (± 0.0) |