# OpenReview forum: "LLMs for Generalizable Language-Conditioned Policy Learning under Minimal Data Requirements"
_ICLR.cc/2025/Conference — Submitted to ICLR 2025_

### Official Review · Reviewer_AZnv · 2024-11-01

**Soundness:** 2
**Presentation:** 2
**Contribution:** 1
**Rating:** 3
**Confidence:** 4

**Summary:**

To train a goal-conditioned autonomous agent, the authors propose four existing main challenges: unlabeled data, limited exploration, unknown data collection policy, and goal generalization. They solve these problems mainly through LLM. In particular, they exploit an LLM to label the unsupervised RL dataset and compress it through an LLM-written program. Once every trajectory is compressed into the latent space, they learn the policy and distillate their information into a pre-trained LLM. The presented method outperforms the baseline, and adequate ablation studies support their intuition.

**Strengths:**

Almost every step of the end-to-end process is automatic. For example, a pre-trained LLM works as a state labeler, so a vast amount of unlabeled data is now usable. Analogously, a state space compressing program is also written through an LLM, which minimizes the human expert's effort. Above all, goal-generalization ability is quite notable. Their experimental results show relatively less performance drop during the test phase compared to the baseline. The ablation study also strongly indicates the capability of hierarchical RL; it learns against a variety of sub-tasks and combines the solutions to reach the ultimate goal.

**Weaknesses:**

The first drawback is its scalability. Their proposed method can be summarized into three main steps: data labeling and compressing, policy training through offline RL, and knowledge distillation. They generate the Python program from the LLM and use that program to extract the minimal information from the original state space. This step totally relies on the characteristics of the environment. In fact, the LLM could not successfully compress the state space unless it was a discrete grid world. I can not imagine how to scale up the LLM-based state compression approach to much more complex environments. In the offline RL step, they used tabular Q-learning. Tabular iteration is one of the most powerful methods to optimize an MDP policy, but its memory consumption hardly limits its scalability. In addition, more strong baselines are required to verify the capability of their method. They mainly compared the vanilla LLM and finetuned LLM, and actually, it is trivial that finetuned LLM significantly outperforms the vanilla LLM. There is also another baseline that works without LLM, but that was published with the benchmark itself in 2018, which is too outdated these days.

**Questions:**

Isn't there any benchmark applicable other than BabyAI? I felt like the entire method is highly correlated to the domain knowledge of the BabyAI benchmark. Even for the BabyAI benchmark, there might be more updated solutions (I am not sure about this. I apologize if not) to be compared. Finally, it is a marginal point, but it would be better to introduce your target task or domain at the beginning of your paper. You keep to using terms such as 'autonomous agent', 'state space', and 'reinforcement learning' and it sounds like your target task is very highly abstracted MDP space. However, you also mention 'natural language-written goals' and discuss how to map them into the neural representation. Maybe introducing your domain and task thoroughly at the beginning of the introduction can enhance its readability.

---

> ### Author Response · Authors · 2024-11-24
> **Response to your review -- part 1**
>
> Thank you for taking the time to review our paper. Below we address the questions and concerned raised.
>
> ---
>
> ### Weaknesses
>
> **W1) Scalability**
>
> **State Abstraction**. Thank you for your comments regarding LLM-induced state abstraction. We would like to clarify that this aspect of the framework is not restricted to grid-based environments. It's the symbolic nature of state representations that makes them particularly suitable for LLM-based abstraction. Any environment state that can be represented in natural language can be summarized by an LLM.
>
> While the focus on symbolic environments may seem limiting, many domains in autonomous agent research naturally offer textual state representations, such as web or computer environments [1,2], video games [3], and multi-turn dialogue systems [4]. Our study of TEDUO in the webshop environment is discussed in detail in the global response. Additionally, most RL environments with tabular representations can be converted into key-value pairs, where values are discretized as needed, a common practice in RL. The abstraction process then focuses on selecting relevant keys.
>
> For pixel-based RL environments, TEDUO could employ vision-language models (VLMs) instead of LLMs for data enhancement and fine-tuning. Prior work has successfully applied LLM-based state abstraction to pixel states [5], where LLMs generate binary masks to occlude irrelevant regions based on the goal. Alternatively, converting pixel-based states into textual representations remains an active research area, as shown in recent studies on the Alfred environment [6].
>
> We believe the provided example of building a state abstraction for environments beyond grid worlds effectively addresses the reviewers' concerns regarding the scalability of TEDUO Step 1.
>
> **UPDATE**: The related work section in the revised manuscript will include the above examples of abstraction functions.
>
> References:
>
> [1] OSWorld: Benchmarking Multimodal Agents for Open-Ended Tasks in Real Computer Environments, 2024
>
> [2] WebShop: Towards Scalable Real-World Web Interaction with Grounded Language Agents, 2022
>
> [3] The NetHack Learning Environment, 2020
>
> [4] LMRL Gym: Benchmarks for Multi-Turn Reinforcement Learning with Language Models, 2023
>
> [5] Learning with Language-Guided State Abstractions, 2024
>
> [6] ALFWorld: Aligning Text and Embodied Environments for Interactive Learning, 2021
>
> **Usage of Tabular Q-learning.** We appreciate the reviewer's concern regarding the scalability of tabular Q-learning. We would like to clarify that step 2 of the TEDUO framework is independent of the specific offline reinforcement learning algorithm used to solve the abstract MDPs generated in step 1. Each abstract MDP is represented by a labeled transition dataset $\mathcal{D}^g$, consisting of tuples $(s^g, a, s'^g, r^g)$, making it compatible with any standard offline RL algorithm. In the implementation presented in Section 5 of our manuscript, we used tabular Q-learning as it was most suitable for the environment under consideration. However, we acknowledge that demonstrating the use of a deep RL method may better illustrate the scalability of the approach. To address this, we include an updated Table 2 (below) with an additional row showing the performance of DQN [1], demonstrating that it can effectively replace tabular Q-learning in our framework. The deep learning models for Q-value estimation combine CNN and dense layers, consistent with the lightweight models used for reward shaping (see Appendix B.3). As shown in Table 2, the performance of DQN is comparable to that of tabular Q-learning.
>
> | Method                       | Success Rate [\%] | Episode Length          | Invalid Actions [\%] |
> |------------------------------|----------------------------|-------------------------|-------------------------------|
> | Step 1 +~GCBC                | 7  (±0.6)      | 474 (±2.3) | 11 (±0.1)        |
> | Steps 1 \& 2 (GCRL- Tabular) | 16 (±0.8)     | 430  (±3.9) | 10 (±0.1)        |
> | Steps 1 \& 2 (GCRL- DQN)     | 15  (±1.2)     | 446  (±5.9) | 15 (±0.1)        |
> | All steps Llama-3-8B         | 65  (±1.4)     | 203  (±6.7) | 21 (±0.7)        |
>
> **UPDATE:** We have updated the manuscript with the above table. Highlighting that TEDUO step 2 is agnostic to the offline RL algorithm used to solve the abstract MDP. Thus, more memory efficient methods can be used in place of tabular Q-learning depending on the application.
>
> Reference:
>
> [1] Playing Atari with Deep Reinforcement Learning, 2013

---

> ### Author Response · Authors · 2024-11-24
> **Response to your review -- part 2**
>
> **W2) Comparions with stronger baselines**
>
> The problem addressed by TEDUO in this paper is novel, and, to the best of our knowledge, no existing methods have been proposed to develop a natural language goal-conditioned policy using the minimal data requirements outlined in the paper (offline, unlabeled, and limited samples). Consequently, part from leveraging the one-shot capabilities of foundation models like large language models (LLMs), there are no established baselines.
>
> We also highlight  that the RL baseline described in Section 5.1 of the manuscript already depends on TEDUO Steps 1 and 2. As no other RL method currently addresses this specific problem, this comparison is not intended as a benchmark for the entire method but rather as a demonstration of the superior generalization capabilities of  Step 3 of TEDUO. To the best of our knowledge, this is the most widely used—and only—RL baseline for training goal-conditioned policies in the BabyAI environment, as evidenced by its use for benchmarking in recent works [1, 2, 3]. If the reviewer is aware of other relevant baselines for training naural-language goal-conditioned policies in BabyAI, we would be happy to include them.
>
> References:
>
> [1] Pre-trained language models for interactive decision-making, 2022
>
> [2] Unified policy for interleaving language reasoning with actions, 2023
>
> [3] Zero-shot compositional policy learning via language grounding, 2023
>
> ### Questions
>
> **Q1) Other benchmarks applicable beyond BabyAI**
>
> Thank you for your comment regarding the applicability of TEDUO to other benchmarks beyond BabyAI. In fact, we discuss the requirements for compatibility and provide examples of other environments that TEDUO is compatible with in Appendix A.4.
>
> Given the significant computational resources and time required to perform all three steps of our pipeline, in particular fine-tuning an LLM agent, our current scope was limited to BabyAI. Nonetheless, the insights derived from this controlled setting are broadly applicable and provide a foundation for future work in environments with similar tabular structures, such as NetHack or Overcooked, which differ mainly in thematic focus (video game dungeon crawling and collaborative cooking, respectively).
>
> Adapting TEDUO to different environments would involve modifying the stages where domain knowledge of the environment is integrated. These include adapting
>
> 1. The prompt for generating the abstraction function (see Appendix B.2)
> 2. The prompt for reward labeling (see Appendix B.3)
> 3. The prompt at inference time (see Appendix B.5)
>
> At these three stages, prior knowledge about the environment's dynamics, the semantics of the instructions, or any information that can warm-start the goal-conditioned policy in Step 3 can be incorporated via natural language.
>
> Moreover, in our global response, we extend the application of TEDUO to a new environment: Webshop. This environment significantly differs from Minigrid in both action and state space. The results on this new benchmark further strengthen our confidence in the method’s applicability across a diverse range of environments.
>
> **Update**: The revised manuscript's Appendix will include a detailed clarification of the domain knowledge integration process within TEDUO.
>
> **Q2) Abstract presentation of the target task**
>
> Thank you for your comment regarding the presentation of our paper. As discussed in Q1), TEDUO is broadly applicable across multiple domains. We have purposefully kept its presentation general and abstract to enable an easy adaptation of the pipeline to other environments and domains beyond the BabyAI grid-world.
>
> ---
>
> We appreciate the time and effort you’ve dedicated to reviewing our paper. We hope that the our answers and proposed changes to the manuscript address your concerns satisfactorily.
>
> Thank you again for helping us improve the quality of this work.
>
> The authors of submission #10568

---

> ### Comment · Reviewer_AZnv · 2024-11-25
> **Comment**
>
> Thank you for your quick response. However, even if TEDUO works fine on BabyAI without tabular RL or statement abstraction, it does not directly imply the scalability of the presented method. The Webshop is also an environment well aligned to a natural language domain, so it limits the significance of the experiment.
>
> Therefore, I will maintain my initial grade.

---

> > ### Author Response · Authors · 2024-11-25
> > **Thank you for your reply**
> >
> > Dear reviewer AZnv,
> >
> > Thank you for your prompt response. We would like to take this opportunity to further clarify our viewpoint.
> >
> > **Symbolic Environments Are Broadly Applicable.** We appreciate the reviewer’s observation regarding symbolic environments and would like to provide additional clarification. While TEDUO's reliance on LLMs aligns it with environments that can be represented symbolically or textually, we do not view this as a significant limitation. Many real-world problems naturally lend themselves to such representations, including domains such as web navigation, software interaction, video games, and multi-turn dialogue systems. Moreover, as discussed in our initial response (W1), advances in vision-language models (VLMs) and textual state abstraction methods are making it increasingly feasible to extend TEDUO’s applicability to pixel-based or more complex environments in future work.
> >
> > **Clarifying the Scope.** To address the reviewer’s concern, we propose to explicitly narrow the scope of our paper in the introduction. We will emphasize that TEDUO is particularly well-suited for symbolic environments while highlighting the potential of VLMs and related approaches as future research directions. This adjustment aims to provide a more precise framing of the current work and its contributions.
> >
> > **Complexity of WebShop.** We would also like to address the reviewer’s concern about the complexity of the WebShop environment.  We argue that the action space in Webshop is far greater than most of typical RL environments. This is due to the following two factors:
> >
> > a) The set of available actions varies dynamically by state, requiring the agent to infer the semantic link between UI elements (i.e., button tooltips) and their underlying actions. Up to 50 distinct buttons can appear on a webpage in a single state.
> >
> > b) The agent must generate natural language inputs (i.e., search queries) to interact with the search bar, effectively expanding the action space to the entirety of natural language.
> >
> > We believe these characteristics distinguish WebShop from more simplistic symbolic benchmarks and underscore its suitability for evaluating the generalization capabilities of TEDUO.
> >
> > **The Value of Our Contributions.** Even if the BabyAI and WebShop environments appear constrained in terms of goal or action diversity, we respectfully argue that the significance of our results lies beyond the complexity of these benchmarks. To the best of our knowledge, TEDUO is the first method to successfully learn generalizable, language-conditioned policies from fully unlabeled data. These benchmarks, while not encompassing all possible complexities, provide a solid foundation for demonstrating TEDUO’s feasibility and robustness. We view these results as critical stepping stones toward scaling instruction-following agents to operate in more diverse and complex environments.
> >
> > **Closing Remarks.** We thank the reviewer for their thoughtful feedback and hope the clarifications above demonstrate the significance and relevance of TEDUO’s contributions. However, if the reviewer maintains their standing that additional environments would better illustrate TEDUO’s impact, we would greatly appreciate concrete examples of environments that TEDUO could be tested on. Specifically, we seek environments that align with the properties outlined in the paper, i.e.:
> >
> > - Supporting a wide range of goal-reaching tasks expressed in natural language.
> > - That do not rely on image or video inputs (due to limited computational resources)
> >
> > Such guidance would be invaluable for extending our work and maximizing its impact in future iterations.
> >
> > ----
> >
> > Thank you for your continued engagement in the rebuttal process.
> >
> > Kind regards,
> >
> > The authors

---

### Official Review · Reviewer_93kq · 2024-11-03

**Soundness:** 2
**Presentation:** 3
**Contribution:** 2
**Rating:** 3
**Confidence:** 3

**Summary:**

This paper introduces TEDUO, a training pipeline aimed at enhancing language-conditioned policy learning in autonomous agents while minimizing data requirements.

TEDUO leverages large language models (LLMs) to address these challenges by employing a three-step process:

**Data Enhancement**: Using LLMs to perform state abstraction and hindsight labeling on an unlabeled dataset of state-action transitions, allowing for the creation of labeled datasets.

**Policy Learning:** Applying offline RL algorithms to learn optimal policies based on the enhanced datasets for a finite set of training goals.

**Generalization**: Fine-tuning a base LLM to distill knowledge about environment dynamics and optimal actions, enabling the model to generalize to previously unseen states and language commands.

**Strengths:**

- The authors discuss a very interesting problem in reinforcement learning: generalization with minimal training data. This is also a general concern for RL.

- Using LLM to augment trajectory data is novel and effective. It has the potential to propose many diverse training data without setting up different environments.

- The performance of TEDUO is effective on BabyAI (table 1) and the authors also explore generalizability to compositional tasks.

**Weaknesses:**

1. The pipeline for data enhancement may rely on the symbolic nature of the tasks. For example, BabyAI is a easy to define symbolic task where changing the shape or color of objects and doors would result in new tasks. However, this would be more difficult to enhance data for more complex environments, e.g. VirtualHome and Alfred. Could the authors provide elaboration on how the pipeline could be generalized to a more complex environment.

2. LLMs are inherently good at simple generalizations (changing only names or objects), however the generalization of reasoning and complex planning are often more challenging. Could the authors benchmark the performance following Table 1 settings on BabyAI more advanced task levels (babyai provides a detailed task levels where some tasks are much more challenging) ? It would be helpful to emphasize what kind of generalization ability does TEDUO possesses.

3. Step 2 is not clearly written. How is an "abstract" MDP solved? For example, for solving a abstract task "picking up the object", what is the solution to the MDP. Also, how to determine if two tasks are the same type of abstract MDP or if they should be labelled as different ?

**Questions:**

Please see questions above.

---

> ### Author Response · Authors · 2024-11-24
> **Response to your review -- part 1**
>
> We thank the reviewer for their insightful comments and constructive feedback. We have carefully considered each point of feedback and provide our point-by-point responses below.
>
> ---
>
> ### Weaknesses
>
> **W1) Focus on symbolic environments**
>
> We acknowledge, as noted in our limitations section, that the proposed instantiation of TEDUO is restricted to environments that can be represented textually. However, we believe this limitation is mitigated by the broad expressiveness of natural language.
>
> 1. Many environments studied in autonomous agent research inherently offer textual state representations, such as web or computer environments [1,2], video games [3], and multi-turn dialogue systems [4].
> 2. For most reinforcement learning environments, which often have tabular representations, these can be converted into key-value pairs. Values can be discretized as needed, a common practice in RL.
> 3. In the case of pixel-based RL (which include both environments proposed by the reviewer), TEDUO could leverage vision-language models (VLMs) instead of LLMs for both the data enhancement and fine-tuning stages. Alternatively, converting pixel-based environments into textual representations is an active area of research, as demonstrated by recent work on the Alfred environment proposed by the reviewer [5].
> 4. In terms of the state abstraction, previous work has applied LLM state abstraction to pixel states [6], with the LLM generating a binary mask occluding the irrelevant areas of the state space with respect to the given goal.
>
> **UPDATE**: The related work section in the revised manuscript will include the above examples of abstraction functions.
>
> [1] OSWorld: Benchmarking Multimodal Agents for Open-Ended Tasks in Real Computer Environments, 2024
>
> [2] WebShop: Towards Scalable Real-World Web Interaction with Grounded Language Agents, 2022
>
> [3] The NetHack Learning Environment, 2020
>
> [4] LMRL Gym: Benchmarks for Multi-Turn Reinforcement Learning with Language Models, 2023
>
> [5] ALFWorld: Aligning Text and Embodied Environments for Interactive Learning, 2021
>
> [6] Learning with Language-Guided State Abstractions, 2024
>
> **W2) Generalization abilities and task complexity**
>
> **Generalization Abilities.** Thank you for your insightful comment on TEDUO's generalization capabilities. The paper demonstrates that TEDUO generalizes across both new environments and new instructions. New environments refer to unseen grid configurations within BabyAI, while new instructions involve goals that are semantically and syntactically distinct from those used during training. Table 1 highlights TEDUO's superior performance compared to standard RL baselines in these scenarios.
>
> We acknowledge the importance of clarifying the scope of TEDUO's generalization. However, uncovering the underlying mechanisms of generalization in deep learning is challenging and remains an open question, particularly for large language models. In the current work, we identify two mechanisms contributing to its success: (1) the fine-tuned LLM's ability to compose previously learned skills to tackle more complex tasks (Section 5.3.1), and (2) the development of core skills for navigation and task completion (Section 5.3.2).
>
> **Benchmarking on More Advanced BabyAI Tasks.** The reviewer raises an excellent point regarding testing TEDUO's performance on more challenging task levels within the BabyAI framework. To clarify, the results in Table 1 were obtained using the *Synth* environment, which is already the most challenging environment type available in BabyAI (ref: [BabyAI documentation](https://minigrid.farama.org/environments/babyai/index.html)). However, we excluded two specific task categories, *Sequence* and *Location*, because they require memory of prior states and thus fall outside the scope of this paper, which focuses on tasks adhering to the Markov property.
>
> **Generalization Beyond BabyAI.** To address the need for evaluating TEDUO in more diverse and complex scenarios, we expanded its application to a significantly different environment, Webshop, as described in our global response. Webshop introduces a larger state and action space, requiring a broader range of generalization capabilities. The results from this new benchmark demonstrate TEDUO's robustness and versatility in handling tasks of varying complexity.

---

> ### Author Response · Authors · 2024-11-24
> **Response to your review -- part 2**
>
> **W3) Clarity of step 2**
>
> We thank the reviewer for highlighting the need for clarification regarding step 2 of our method. We propose the following clarification to summarize step 2 :
>
> At the end of step 1, we generate a collection of goal-conditioned abstract MDPs”, which are synthetically generated MDPs. Each abstract MDP, denoted as $\mathcal{M}^g$, is materialized by a labeled transition dataset  $\mathcal{D}^g$ , consisting of tuples $(s^g, a, s'^g, r^g)$. Individually, each of these MDPs constitutes a standard offline RL problem. In step 2, we independently solve each of these problems using a standard offline RL algorithm. For this specific instantiation of TEDUO, we utilize Tabular Q-learning, as it is well-suited for the environments considered. However, other RL methods could be employed. Here, by solving an abstract MDP we mean that for each abstract MDP corresponding to a training goal in $g \in \mathcal{G}^{tr}$, we derive learn the optimal policy based on the corresponding transition dataset $\mathcal{D}^g$ and the chosen offline RL algorithm. As a result, step 2 produces a set of collection of policies,  $\\{\pi^g\\}_{g \in \mathcal{G}^{tr}}$, which are then ready for teaching to a pre-trained LLM in step 3. **UPDATE:** Based on your feedback, we will refine the explanation of TEDUO Step 2 for clarity in the revised manuscript.
>
> ---
>
> Once again, thank you for your thoughtful comments and questions. We hope that our responses and the resulting updates have addressed your concerns satisfactorily.
>
> Thank you for helping us improve the quality of our submission.
>
> King regards,
>
> The authors of submission #10568

---

> ### Comment · Reviewer_93kq · 2024-11-24
> **Thank you for your response**
>
> Thank you for your response and for the effort to introduce a more complex environment like WebShop.
>
> However, in my observation, WebShop still appears to be a relatively symbolic environment with limited diversity and a constrained action space. For example, all tasks are primarily focused on browsing objects, with actions mainly limited to search [A] and buy [A]. While I sincerely appreciate the authors' efforts to incorporate more complex tasks, I believe that a truly diverse benchmark would require a broader variety of tasks and a significantly larger action space (involving more than just a few actions).
>
> It is not entirely clear to me how TEDUO could be directly applied in such cases, as defining and solving each type of problem may become substantially more challenging. Moreover, the experience gained from scaling in this environment may not be easily transferable to more diverse and complex settings.
>
> Therefore, I would like to maintain my initial rating but hope that the authors consider including or adapting to such tasks in the future.
>
> Additional question:
>
> - Could the authors provide the row number where they provide abstraction functions? I didn't find any in the related work section in the main paper or the appendix.

---

> ### Author Response · Authors · 2024-11-25
> **Thank you for your reply**
>
> Dear reviewer 93kq,
>
> Thank you for your prompt response. We would like to take this opportunity to further clarify our viewpoint.
>
> **Symbolic Environments Are Broadly Applicable.** We appreciate the reviewer’s observation regarding symbolic environments and would like to provide additional clarification. While TEDUO's reliance on LLMs aligns it with environments that can be represented symbolically or textually, we do not view this as a significant limitation. Many real-world problems naturally lend themselves to such representations, including domains such as web navigation, software interaction, video games, and multi-turn dialogue systems. Moreover, as discussed in our initial response (W1), advances in vision-language models (VLMs) and textual state abstraction methods are making it increasingly feasible to extend TEDUO’s applicability to pixel-based or more complex environments in future work.
>
> **Clarifying the Scope.** To address the reviewer’s concern, we propose to explicitly narrow the scope of our paper in the introduction. We will emphasize that TEDUO is particularly well-suited for symbolic environments while highlighting the potential of VLMs and related approaches as future research directions. This adjustment aims to provide a more precise framing of the current work and its contributions.
>
> **Complexity of WebShop.** We would also like to address the reviewer’s concern about the complexity of the WebShop environment.  We argue that the action space in Webshop is far greater than most of typical RL environments. This is due to the following two factors:
>
> a) The set of available actions varies dynamically by state, requiring the agent to infer the semantic link between UI elements (i.e., button tooltips) and their underlying actions. Up to 50 distinct buttons can appear on a webpage in a single state.
>
> b) The agent must generate natural language inputs (i.e., search queries) to interact with the search bar, effectively expanding the action space to the entirety of natural language.
>
> We believe these characteristics distinguish WebShop from more simplistic symbolic benchmarks and underscore its suitability for evaluating the generalization capabilities of TEDUO.
>
> **The Value of Our Contributions.** Even if the BabyAI and WebShop environments appear constrained in terms of goal or action diversity, we respectfully argue that the significance of our results lies beyond the complexity of these benchmarks. To the best of our knowledge, TEDUO is the first method to successfully learn generalizable, language-conditioned policies from fully unlabeled data. These benchmarks, while not encompassing all possible complexities, provide a solid foundation for demonstrating TEDUO’s feasibility and robustness. We view these results as critical stepping stones toward scaling instruction-following agents to operate in more diverse and complex environments.
>
> **Closing Remarks.** We thank the reviewer for their thoughtful feedback and hope the clarifications above demonstrate the significance and relevance of TEDUO’s contributions. However, if the reviewer maintains their standing that additional environments would better illustrate TEDUO’s impact, we would greatly appreciate concrete examples of environments that TEDUO could be tested on. Specifically, we seek environments that align with the properties outlined in the paper, i.e.:
>
> - Supporting a wide range of goal-reaching tasks expressed in natural language.
> - That do not rely on image or video inputs (due to limited computational resources)
>
> Such guidance would be invaluable for extending our work and maximizing its impact in future iterations.
>
> ----
>
> **Answer to your question.** Explanations regarding the implementation of the abstraction function can be found in Appendix B.2. For a complete description, the code related to the instantiation of TEDUO presented in the paper is provided in the supplementary material.
>
> ---
>
> Thank you for your continued engagement in the rebuttal process.
>
> Kind regards,
>
> The authors

---

### Official Review · Reviewer_3cHE · 2024-11-04

**Soundness:** 3
**Presentation:** 3
**Contribution:** 2
**Rating:** 6
**Confidence:** 3

**Summary:**

The paper presents a novel approach, TEDUO, for training autonomous agents. TEDUO addresses the challenge of training agents that can generalize to new goals and states while requiring minimal labeled data. The approach leverages large language models (LLMs) as data enhancers and policy generalizers, which allows the use of easy-to-obtain, unlabeled datasets. The approach uses the knowledge of pre-trained large language models to achieve generalization ability, and experiments show that the method has better results and generalization ability than the baseline. However, there are some limitations: First, the method relies on the knowledge of pre-trained large language models, and if the capability of large language models is not good enough, the effect of the method may be significantly decreased. In addition, the experimental scene is relatively simple. Although the experiment proves that the method can generalize, it may not have enough generalization ability for the complex scene in reality.

**Strengths:**

1) TEDUO effectively leverages LLMs as both data enhancers and generalizers, thus reducing the need for expensive labeled data.
2) Focus on unlabeled, pre-collected data sets with fewer assumptions about their quality or origin.
3) Experiments show that TEDUO proves its ability to improve and generalize compared to the baseline and achieves better performance in zero-shot Settings, making it suitable for new scenarios.

**Weaknesses:**

1) The paper assumes that environment states can be adequately represented textually. This may restrict the approach's applicability to more complex, real-world scenarios where high-dimensional or continuous representations are required.
2) The success of the TEDUO framework heavily relies on the LLMs' pre-trained knowledge. If the domain is not well-represented in the LLM's training data, performance may suffer, limiting generalizability to specialized fields.
3) The focus on simpler environments like BabyAI may not reflect the challenges of real-world applications, where environments can be more dynamic and less predictable.
4) The approach introduces significant computational demands, especially for LLM-based state abstraction and generalization.

**Questions:**

1) While the method shows generalization capabilities, the extent to which it can handle various novel goals and states in practice remains unclear. More thorough testing in diverse scenarios is needed.
2) Performance on small models may be poor.

---

> ### Author Response · Authors · 2024-11-24
> **Response to your review -- part 1**
>
> Thank you for taking the time to review our work. We appreciate the reviewer’s detailed feedback on our submission and we address their concerns point by point in the below response.
>
> ---
>
> ### Weaknesses
>
> **W1)** **Limitation to symbolic and tabular environments**
>
> We acknowledge, as noted in our limitations section, that the proposed instantiation of TEDUO is restricted to environments that can be represented textually. However, we believe this limitation is mitigated by the broad expressiveness of natural language.
>
> 1. Many environments studied in autonomous agent research inherently offer textual state representations, such as web or computer environments [1,2], video games [3], and multi-turn dialogue systems [4].
> 2. For most reinforcement learning environments, which often have tabular representations, these can be converted into key-value pairs. Values can be discretized as needed, a common practice in RL.
> 3. In the case of pixel-based RL, TEDUO could leverage vision-language models (VLMs) instead of LLMs for both the data enhancement and fine-tuning stages. Alternatively, converting pixel-based environments into textual representations is an active area of research, as demonstrated by recent work [5].
>
> References:
>
> [1] OSWorld: Benchmarking Multimodal Agents for Open-Ended Tasks in Real Computer Environments, 2024
>
> [2] WebShop: Towards Scalable Real-World Web Interaction with Grounded Language Agents, 2022
>
> [3] The NetHack Learning Environment, 2020
>
> [4] LMRL Gym: Benchmarks for Multi-Turn Reinforcement Learning with Language Models, 2023
>
> [5] ALFWorld: Aligning Text and Embodied Environments for Interactive Learning, 2021
>
> Given the above points, we believe that focusing our attention on environments representable in a text format is not a major limitation of the proposed approach.
>
> **W2) Success relies on the LLMs’ pre-trained knowledge**
>
> In our Limitations section, we acknowledged that a key assumption of our work is that LLMs pretrained on internet-scale data may serve as a valuable source of knowledge to automate data labelling and state abstraction. Nevertheless, this assumption is widely supported in the RL community, as evidenced by numerous studies leveraging LLMs in diverse domains, such as video games [1], household tasks [2], and robotics [3]. Moreover, as LLMs continue to advance and expand their knowledge across more domains, this assumption should become increasingly valid.
>
> References:
>
> [1] Motif: Intrinsic Motivation from Artificial Intelligence Feedback, 2023
>
> [2] Guiding Pretraining in Reinforcement Learning with Large Language Models, 2023
>
> [3] Language to Rewards for Robotic Skill Synthesis, 2023
>
> **W3) Focus on simple environments**
>
> We appreciate the reviewer's encouragement to provide an additional benchmarking environment for TEDUO. To address this point, we performed a new set of experiments in a new environment: Webshop. The details and results are provided in the global response.
>
> **W4) Increased computational demands**
>
> We acknowledge the reviewer’s concern regarding the computational cost of TEDUO. However, we emphasize that one of its primary objectives is to utilize computational resources—by extracting knowledge from pre-trained LLMs—to reduce the need for additional data samples (abstraction stage) and human labeling (automatic labeling stage), which can often be prohibitively expensive or infeasible, depending on the use case.
>
> Moreover, each stage of the method is designed to minimize computational overhead. For instance, the abstraction function generates a Python function for each goal instead of directly applying the LLM to individual transitions (see Section 3.1.1). In the reward labeling stage, we improve upon prior methods that label every transition [1] or datasets of pairs [2], by labeling only a subset of the data and training a lightweight goal-detection model (see Section 3.1.2). Finally, we exploit the deterministic nature of the environment to train the model to predict the next *N* actions (with *N* ≫ 1), significantly reducing training and inference costs (see Section 3.3).
>
> To the best of our knowledge, this is the first pipeline capable of generating instruction-following policies with such minimal data requirements. As highlighted in section 5.4, our approach shifts the bottleneck from the limited availability of real-world observational, labelled data to computational power.

---

> ### Author Response · Authors · 2024-11-24
> **Response to your review -- part 2**
>
> ### Questions
>
> **Q1) The extent of generalisation to new goals and states**
>
> As detailed in the global response, the new Webshop environment differs substantially from Minigrid in both action and state spaces. We believe that applying TEDUO to this novel setting reinforces its generalization capabilities, addressing the reviewer’s concerns.
>
> **Q2) Performance on small models**
>
> To address the reviewer’s question about the performance of our methods with smaller LLMs, we present two analyses reflecting the two roles of LLMs in our pipeline: **data enhancer** and **generalizer**.
>
> **Data Enhancer**
>
> In applying TEDUO to the webshop environment, we tested three different model sizes for reward labeling during TEDUO step 1. The table below presents a comparison between the synthetically generated rewards and the true rewards provided by the webshop environment. We evaluated the performance using two metrics: Mean Squared Error (MSE) and accuracy, calculated by thresholding both rewards at 0.5.
>
> | **Models** | **MSE** | **Accuracy** |
> | --- | --- | --- |
> | Llama-3.2-1B-Instruct | 9.8e-2 | 70% |
> | Llama-3-8B-Instruct | 9.1e-2 | 73% |
> | Llama-3-70B-Instruct | 8.9e-2 | 68% |
>
> The MSE results suggest that larger models align more closely with ground truth rewards, though the differences are minimal. Accuracy indicates that for this task, model size ( beyond 1B parameters) has limited impact.
>
> **Generalization Performance**
>
> We conducted supervised fine-tuning (TEDUO step 3) on the same datasets used for the paper benchmarking, substituting Llama-3-8B-Instruct by Llama-3.2-1B-Instruct. Both fine-tuned models were evaluated under the paper benchmark condition, with results summarized below:
>
> | **Method** | **Environment** | **Goals** | **Success Rate** | **Episode Length** | **Invalid Actions** |
> | --- | --- | --- | --- | --- | --- |
> | **TEDUO-Llama-3-1B** | train | train | 27 (±2.5) | 375 (±11.9) | 48 (±2.2) |
> |  | test | train | 19 (±1.9) | 410 (±9.2) | 46 (±2.0) |
> |  | train | test | 34 (±2.2) | 341 (±10.1) | 38 (±2.0) |
> |  | test | test | 33 (±2.3) | 343 (±10.8) | 37 (±1.9) |
> | **TEDUO-Llama-3-8B** | train | train | 65 (±1.4) | 203 (±6.7) | 21 (±0.7) |
> |  | test | train | 53 (±1.1) | 257 (±5.4) | 27 (±0.7) |
> |  | train | test | 55 (±1.6) | 241 (±7.5) | 22 (±1.1) |
> |  | test | test | 45 (±1.3) | 286 (±6.1) | 31 (±1.2) |
>
> For the generalization role, the number of parameters in the LLM significantly impacts performance. We observe a 12% decrease in success rate under the most challenging setting (new goals and new environments). Notably, this decline is more pronounced for training goals, indicating limited memorization capabilities in the 1B-parameter models. However, despite this, the generalization performance of these smaller fine-tuned models remains well above the baseline presented in the paper. These findings confirm that smaller models perform worse as expected, but they can still effectively generalize learned policies to new goals and environments.
>
> ### Updated manuscript
>
> These additional comparisons and analyses will be made available in the Appendix of the revised manuscript.
>
> ---
>
> Once again, thank you for your thoughtful comments and questions. We hope that our responses and the resulting updates have addressed your concerns satisfactorily.
>
> Thank you for helping us improve the quality of our submission.
>
> King regard,
> The authors of submission #10568

---

> ### Comment · Reviewer_3cHE · 2024-11-26
>
> Thank you for your elaboration. For the generalization of the proposed method, I think the cases raised by the authors still lack complexity, e.g. webshop. So it still doesn't address my concern.
> Therefore, I will keep my rating.

---

### Official Review · Reviewer_bzcZ · 2024-11-04

**Soundness:** 3
**Presentation:** 4
**Contribution:** 3
**Rating:** 5
**Confidence:** 4

**Summary:**

This paper proposes a policy learning method under minimal data requirements, enabling LLMs to act as both cheap data enhancers and flexible generalizers. This approach addresses the traditional RL methods' dependency on large amounts of offline data or real-time interactions. Experimental results confirm the effectiveness of the method and demonstrate a scaling phenomenon between the method and computational resources.

**Strengths:**

- The paper uses a substantial amount of symbolic notation to make the descriptions clearer.
- The method proposed in this paper addresses the need for real-time interaction feedback or large amounts of offline data.
- The paper demonstrates the potential performance of the method when provided with greater computational power.

**Weaknesses:**

1. The pipeline proposed in this paper requires multiple stages, which imposes certain demands on computational resources.
1. Even though the paper provides an explanation for selecting BabyAI as the sole benchmark, relying on a single benchmark to validate the method's effectiveness is insufficient. Given the claimed general applicability of the method, more robust and diverse experimental results should be presented to strengthen the evidence.

**Questions:**

1. Following Weakness 2, is it possible for this method to exhibit performance generalization across different datasets?
1. It appears that Figure 1 is not referenced in the main text.
1. Line 1220, the citation is not displayed correctly.

---

> ### Author Response · Authors · 2024-11-24
> **Global Response**
>
> We would like to express our gratitude to all the reviewers for their constructive feedback and insights on our submission. In the below response we summarise and address the common feedback of the reviewers. Remaining questions and concerns have been answered in individual responses posted under each review.
>
> ---
>
> ### Single environment evaluation
>
> We identified and acknowledge the reviewers’ common concern regarding the benchmarking our method on a single environment. However, in goal-conditioned RL, solving multiple goals across various environments incurs significant implementation and computational costs. We argue that a detailed analysis within one environment, including ablation studies and independent evaluations of every component, provides deeper insights into the method's potential and inner-working in comparison to a shallower evaluation across many environments sharing similar structures. This approach has been adopted by several recognised prior studies:
>
> - Voyager: An Open-Ended Embodied Agent with Large Language Models, TMLR, 2024
> - Human-Timescale Adaptation in an Open-Ended Task Space, ICML, 2023
> - Discovering Hierarchical Achievements in Reinforcement Learning via Contrastive Learning, Neurips, 2023
> - ELLA: Exploration through Learned Language Abstraction, Neurips, 2023
> - Grounding Large Language Models in Interactive Environments with Online Reinforcement Learning, ICML, 2023
> - SPRING: Studying the Paper and Reasoning to Play Games, Neurips, 2023
> - Learning to Understand Goal Specifications by Modelling Reward, ICLR, 2019
>
> ### UPDATE: New experiments on Webshop
>
> Nonetheless, to address the reviewer's concerns, we include an additional benchmark of TEDUO in the **Webshop** environment [1]. Webshop is a simulated e-commerce environment where an agent given product requirements must locate corresponding items by navigating a website. This environment is particularly relevant due to its dynamically evolving action space, presenting two key challenges:
>
> 1. **Dynamic UI Actions:** The available actions vary by state due to changes in the website's UI elements. We address this challenge by concatenating the available actions with the state description.
> 2. **Linguistic Action Requirements:** The agent must generate linguistic inputs (e.g., search queries) to use the search bar. This highlights the necessity for the fine-tuned LLM to avoid catastrophic forgetting [2] and produce coherent keywords for narrowing searches.
>
> **State Representation and Abstraction Function:** In Webshop, the initial state is represented as the HTML code of the webpage. However, akin to our abstraction function, the authors propose a simplified representation retaining only relevant text and interactive UI elements, excluding HTML formalism. While we adopt this representation due to time constraints for the rebuttal, we are confident that TEDUO Step 1 could replicate an abstraction of this kind given the demonstrated ability of LLMs to parse HTML [3]. Additionally, as Webshop is non-Markovian, we ensure Markovian properties by using historical states--concatenating all states from the initial to the current state. References to "states" in the following discussion refer to these history-augmented states.
>
> **Experimental Setup:** We collected 5,000 unlabeled trajectories across 1,500 instructions using a data collection policy that randomly selects actions and prompts an LLM (Llama-3-8B-Instruct) to generate search bar keywords. In TEDUO Step 1, goal-conditioned reward labeling was performed by prompting an LLM to score the fit between the instruction and the purchased product across four criteria: category, attributes, options, and price (these are the 4 criteria that impact the true reward function proposed by the environment). The average of these scores served as the synthetic reward. Following the original paper's approach, TEDUO Step 2 employed Tabular Q-Learning, and Step 3 fine-tuned Llama-3-8B-Instruct using optimal trajectories derived from Step 2.

---

> ### Author Response · Authors · 2024-11-24
> **Global Response (continued)**
>
> **Results:** The results averaged over 100 instructions are shown below. Unlike Minigrid, Webshop does not feature multiple environment types, prohibiting generalization evaluation across environments. Baselines include general-purpose LLMs with the ReAct prompting technique [1], as it is state-of-the-art in our low-data settings for this environment. Vanilla prompting was excluded as it failed to reach the purchasing step within the step limit (15) and thus did not achieve valid rewards. The scores represent the true environment reward (scaled to [0, 100]).
>
> | Method | Goals | Score | Episode Length |
> | --- | --- | --- | --- |
> | ReAct-Llama-3-8B | Training/Testing | 8.4 ± 0.8 | 14.1 ± 0.1 |
> | ReAct-Llama-3-70B | Training/Testing | 13.8 ± 0.9 | 14.2 ± 0.1 |
> | TEDUO-Llama-3-8B | Training | 34.6 ± 1.3 | 3.9 ± 0.1 |
> | TEDUO-Llama-3-8B | Testing | 39.1 ± 2.4 | 4.0 ± 0.2 |
>
> This benchmark demonstrates that TEDUO successfully learns from unlabeled trajectories in the Webshop environment. TEDUO significantly outperforms the ReAct prompting approach, particularly when applied to smaller models (compared to GPT-3.5 in the original paper). While ReAct prompting provides some improvement over standard prompting, its performance remains limited, whereas TEDUO achieves substantially better results.
>
> **Ablation study:** Similar to the approach in the paper, we provide an ablation study below that summarizes the performance improvements achieved after each TEDUO step. The results demonstrate that TEDUO Steps 1 & 2 effectively produce improved policies over the data collection policy for the training goals. Furthermore, TEDUO Step 3 successfully achieves its generalization objective, extending the learned policy's performance to new instructions.
>
> | Method | Goals | Score | Episode Length |
> | --- | --- | --- | --- |
> | Data collection policy (random) | Training/Testing | 5.6 ± 0.5 | 12.3 ± 0.2 |
> | Step 1 & 2 (GCRL) | Training | 37.7 ± 0.9 | 5.9 ± 0.2 |
> | TEDUO-Llama-3-8B | Training | 34.6 ± 1.3 | 3.9 ± 0.1 |
> | TEDUO-Llama-3-8B | Testing | 39.1 ± 2.4 | 4.0 ± 0.2 |
>
> ### Updated manuscript
>
> This additional benchmark and analysis will bee included as an Appendix in the revised manuscript.
>
> [1] WebShop: Towards Scalable Real-World Web Interaction with Grounded Language Agents, 2023
>
> [2] An Empirical Study of Catastrophic Forgetting in Large Language Models During Continual Fine-tuning, 2024
>
> [3] Understanding HTML with Large Language Models, 2023
>
> ---
>
> We are grateful for the reviewers' feedback, which helped us improve the presentation of our work. We are open to further discussions to clarify any aspects of our submission.
>
> Kind regards,
>
> The authors of submission #10568

---

> ### Author Response · Authors · 2024-11-29
> **Response to your feedback (reviewer bzcZ)**
>
> Dear reviewer bzcZ,
>
> We sincerely apologize for the oversight in addressing your specific comment earlier. Due to a mix-up, we inadvertently posted a global response that encompassed our thoughts on related points, but we failed to follow up with an individual reply to your detailed feedback. We apologize for this delay and greatly appreciate your patience.
> Below, we provide a direct response to your comment:
>
> ---
>
> ### Weaknesses
>
> **W1) Multiple stages demand computations resources**
>
> We acknowledge the reviewer’s concern regarding the computational cost of TEDUO. However, we emphasize that one of its primary objectives is to utilize computational resources—by extracting knowledge from pre-trained LLMs—to reduce the need for additional data samples (abstraction stage) and human labeling (automatic labeling stage), which can often be prohibitively expensive or infeasible, depending on the use case.
>
> Moreover, each stage of the method is designed to minimize computational overhead. For instance, the abstraction function generates a Python function for each goal instead of directly applying the LLM to individual transitions (see Section 3.1.1). In the reward labeling stage, we improve upon prior methods that label every transition [1] or datasets of pairs [2], by labeling only a subset of the data and training a lightweight goal-detection model (see Section 3.1.2). Finally, we exploit the deterministic nature of the environment to train the model to predict the next *N* actions (with *N* ≫ 1), significantly reducing training and inference costs (see Section 3.3).
>
> To the best of our knowledge, this is the first pipeline capable of generating instruction-following policies with such minimal data requirements. As highlighted in section 5.4, our approach shifts the bottleneck from limited availability of real-world observational, labelled data to computational power.
>
> **W2) Experimental evaluation limited to BabyAI**
>
> We appreciate the reviewer's encouragement to provide an additional benchmarking environment for TEDUO. To address this point, which may also interest other reviewers, we have analyzed a new use case: Webshop. The details and results are provided in the global response.
>
> ### Questions
>
> **Q1) Is it possible for this method to exhibit performance generalization across different datasets?**
>
> Thank you for asking this question. Indeed, this is one of the major advantages of distilling knowledge of the optimal policies into an LLM, as it could enable knowledge sharing across multiple datasets, potentially with varying action and state spaces. In fact, the experiment of section 5.3.1 can be seen as a simple illustration of this, where two datasets: one collected from environment A and one collected from environment B, are used to train the language-conditioned agent that is capable of generalising to the third environment C. Assessment of more complex scenarios of knowledge sharing across datasets collected in environments with varying state or action spaces is an interesting avenue for future work. **UPDATE:** Following your question, we have included a note on this aspect in the Discussion section of our paper.
>
> **Q2) and Q3).** Thank you for your attention to detail. **UPDATE:** We have now added a reference to Figure 1 in the Introduction section of the paper and updated the citation in line 1220.
>
> **References**
>
> [1] Reward Design with Language Models, 2023
>
> [2] Motif: Intrinsic Motivation from Artificial Intelligence Feedback, 2024
>
> ---
>
> Thank you for raising these important points. We value your time and insights, and we hope this response provides the necessary clarification. Please feel free to let us know if you have any further questions or concerns. Once again, we apologise for our initial oversight.
>
> Kind regards,
>
> The authors of submission #10568

---

### Meta-Review · Area_Chair_HnfH · 2024-12-21

**Metareview:**

This paper proposes a policy learning method under minimal data requirements, enabling LLMs to act as both cheap data enhancers and flexible generalizers. This approach addresses the traditional RL methods' dependency on large amounts of offline data or real-time interactions. Experimental results confirm the effectiveness of the method and demonstrate a scaling phenomenon between the method and computational resources.

While the paper provided an interesting set of results, reviewers were primarily concerned about the scalability and difficulty results considered in the paper. The authors are encouraged to provided results on more complex environments such as the harder settings of BabyAI.

**Additional Comments On Reviewer Discussion:**

The authors and reviewers had an extended discussion but the reviewers were ultimately not convinced by the difficulty of tasks shown by the authors.

---

### Decision · Program_Chairs · 2025-01-22

Reject